# SOX21 modulates SOX2-initiated differentiation of epithelial cells in the extrapulmonary airways

Evelien Eenjes[1], Marjon Buscop-van Kempen[1], Anne Boerema-de Munck[1], Gabriela G Edel[1], Floor Benthem[1], Lisette de Kreij-de Bruin[1], Marco Schnater[1], Dick Tibboel[1], Jennifer Collins[1], Robbert J Rottier[1,2]*

[1]Department of Pediatric Surgery, Erasmus Medical Center – Sophia Children's Hospital, Rotterdam, Netherlands; [2]Department of Cell biology, Erasmus Medical Center, Rotterdam, Netherlands

**Abstract** SOX2 expression levels are crucial for the balance between maintenance and differentiation of airway progenitor cells during development and regeneration. Here, we describe patterning of the mouse proximal airway epithelium by SOX21, which coincides with high levels of SOX2 during development. Airway progenitor cells in this SOX2+/SOX21+ zone show differentiation to basal cells, specifying cells for the extrapulmonary airways. Loss of SOX21 showed an increased differentiation of SOX2+ progenitor cells to basal and ciliated cells during mouse lung development. We propose a mechanism where SOX21 inhibits differentiation of airway progenitors by antagonizing SOX2-induced expression of specific genes involved in airway differentiation. Additionally, in the adult tracheal epithelium, SOX21 inhibits basal to ciliated cell differentiation. This suppressing function of SOX21 on differentiation contrasts SOX2, which mainly drives differentiation of epithelial cells during development and regeneration after injury. Furthermore, using human fetal lung organoids and adult bronchial epithelial cells, we show that SOX2+/SOX21+ regionalization is conserved. Lastly, we show that the interplay between SOX2 and SOX21 is context and concentration dependent leading to regulation of differentiation of the airway epithelium.

*For correspondence:
r.rottier@erasmusmc.nl

Competing interests: The authors declare that no competing interests exist.

## Introduction

The lungs are composed of a highly branched tubular system of airways lined with specialized cell types, which together moisten and warm the air and filter out inhaled substances. The alveoli are located at the distal end of the airways. They consist of a thin layer of epithelium encircled by a network of blood vessels to facilitate an exchange of oxygen and carbon dioxide. During lung development, growth and differentiation along the proximal-distal axis occur simultaneously. A well-regulated balance between progenitor cell maintenance, proliferation, and differentiation is essential to ensure fully functional lungs at birth.

The formation of lung endoderm starts from the ventral anterior foregut by the development of two lung buds, forming the left and right lung. As the growing lung buds expand, the future trachea is separated from the esophagus proximally of the lung buds. Through a repetitive process of branching of the growing tips and outgrowth of the newly formed branches, a complex bronchial tree develops (*Metzger et al., 2008*). Regionalization of the branching structures occurs along the proximal-distal axis. Distal progenitors expressing the SRY-box protein SOX9 and the HLH protein ID2 generate new branches and will ultimately give rise to alveolar cells (*Rawlins et al., 2009a*). While the buds grow and expand, SOX9+ progenitor cells gradually become more distant from the branch-inducing FGF10 signal secreted by the distal mesenchymal cells, resulting in loss of SOX9

expression and initiation of SOX2 expression (*Park et al., 1998*; *Weaver et al., 1999*). These SOX2+ progenitor cells mark the non-branching epithelium and give rise to the airway lineages.

SOX2 is a critical transcription factor in the development of the airway and epithelial lineages. SOX2 deficiency results in aberrant tracheobronchial epithelium due to the loss of basal cells (*Rock et al., 2009*; *Tompkins et al., 2009*; *Wang et al., 2013*). In contrast, SOX2 overexpression results in increased basal cell numbers and perturbed branching morphogenesis (*Gontan et al., 2008*; *Ochieng et al., 2014*). The necessity of proper SOX2 levels in airway development is therefore well documented, but it remains unclear how the balance between SOX2+ progenitor maintenance and early cell fate determination is regulated.

Previously, we showed that increased SOX2 expression during mouse lung development leads to increased SOX21 expression in airway epithelium (*Gontan et al., 2008*). While the function of SOX21 in lung development is not known, SOX21 was found to be enriched in SOX2+ stalk (differentiating bronchiole) versus tip progenitor cells of the human fetal lung (*Nikolić et al., 2017*). Concomitant expression of SOX2 and SOX21 results in either a synergistic or antagonistic function depending on the tissue or environmental stimuli (*Mallanna et al., 2010*; *Freeman and Daudet, 2012*; *Whittington et al., 2015*; *Goolam et al., 2016*).

Here, we hypothesized that SOX21 is an important regulator of SOX2+ progenitor maintenance during development and regeneration of airway epithelium. In this study, we demonstrate that during lung development SOX21 marks a region in the proximal-distal patterning of the airway tree. SOX21 defines the epithelium of the airways where SOX2+ progenitor cells differentiate into basal cells. Using SOX2 and SOX21 mutant mice, we show that SOX2 and SOX21 have opposite functions, as SOX21 suppresses and SOX2 promotes the differentiation of airway progenitor cells during lung development. In contrast, transgenic mice with high ectopic expression of either SOX2 or SOX21 during lung development presented both airway branching defects and increased basal cell numbers. In mouse adult lungs, co-expression of SOX2 and SOX21 is maintained in the epithelium of the extrapulmonary airways (trachea and main bronchi), the region where basal cells reside. Adult mouse basal cells with reduced SOX21 levels are more prone to differentiate to ciliated cells, while reduced SOX2 levels decreased differentiation towards ciliated cells. This opposing effect on differentiation is similar to what we observed in airway progenitor cells during lung development. Finally, SOX2 and SOX21 have a conserved expression pattern and function in human airway epithelial cells. In human air-liquid interface (ALI) cultures, both the reduction of SOX21 or SOX2 levels resulted in an increased differentiation to ciliated cells. Taken together, our results demonstrate that the balance of SOX2 and SOX21 expression regulates airway progenitor cell maintenance and differentiation during lung development and regeneration.

## Results

### SOX2 and SOX21 regionalize the extrapulmonary airways in a proximal–distal pattern during branching morphogenesis

To gain more insight into the interplay between SOX2 and SOX21, we first examined the spatial and temporal distribution of SOX21 in mouse lung epithelium with respect to the SOX2–SOX9 proximal–distal patterning at different stages of lung development.

The earliest expression of SOX21 during lung development was found at gestational age E11.5. A few SOX21-expressing cells were found in the most proximal region of the SOX2+ epithelium (*Figure 1A*). SOX2 expression, which can be detected at E9.5, therefore precedes SOX21 (*Que et al., 2007*; *Gontan et al., 2008*). At E11.5, abundant SOX21 expression was seen in cells of the thymus and esophagus, which also co-expressed SOX2. At E12.5, SOX21 expression became more apparent, but stayed restricted to the proximal region of the SOX2+ epithelium, the trachea, and main bronchi. From E13.5 onwards, SOX21 was expressed throughout the trachea and main bronchi, but was absent in the smaller SOX2+ airways. Labeling from proximal to distal, four zones could be distinguished in the developing lung epithelium (*Figure 1A*): zone 1, the most proximal zone, comprised the developing trachea and main bronchi, which contain SOX2+SOX21+ airway epithelial cells; zone 2, which contains only SOX2+ airway epithelial cells; zone 3, a transition zone, in which distal SOX9+ progenitors transition into SOX2+ airway progenitors (*Mahoney et al., 2014*); and zone 4, the most distal part of the lung epithelium containing SOX9+ progenitors

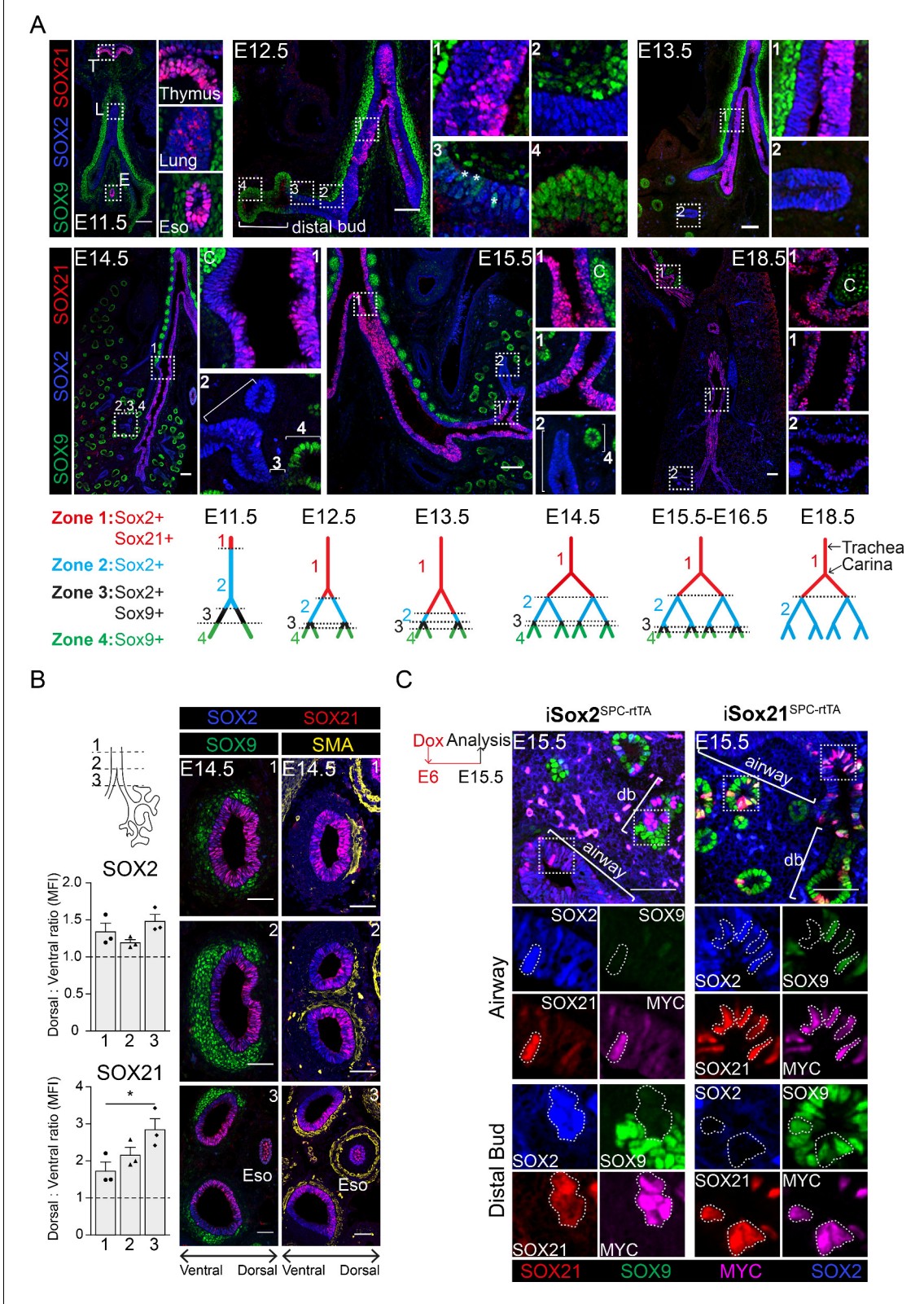

**Figure 1.** SOX21 is expressed in the proximal region of the SOX2+ non-branching zone of the airway epithelium. (**A**) Co-staining of SOX9 (green) for distal buds, SOX2 (blue) for proximal epithelium, and SOX21 (red) at different stages of lung development. Boxed areas are sown as enlarged inserts. Schematic representations show the distribution of SOX9, SOX2, and SOX21 in the different identified zones in the branching airways during gestation. At E12.5, the asterisks indicate the cells in zone three expressing both SOX2 and SOX9. T = thymus, L = lung, E = esophagus, and C = cartilage. Scale

*Figure 1 continued on next page*

*Figure 1 continued*

bar = 50 µm. (**B**) Transversal sections at three different locations of an E14.5 trachea and main bronchi stained with SOX2 (blue), SOX9 (green), SOX21 (red), and smooth muscle actin (SMA) (yellow). Location 1 = trachea, 2 = around the carina, and 3 = immediately distal of the carina. SOX9+ mesenchymal cells surround the ventral side of the trachea and bronchi and SMA+ cells surround the dorsal side. The graphs show the ratio of mean fluorescence intensity (MFI) between ventral and dorsal of SOX2 and SOX21 at the three different locations. Scale bar = 50 µm. Eso = esophagus. (**C**) Sections of E15.5 lungs of doxycycline induced iSox2$^{SPC-rtTA}$ and iSox21$^{SPC-rtTA}$. Expression of transgenic *Myc*-tagged *Sox2* or *Sox21* was induced by giving doxycycline from E6 onwards. Sections are stained with SOX2 (blue), SOX9 (green), SOX21 (red), and MYC (purple). Scale bar = 50 µm. Db = distal bud.

The online version of this article includes the following source data and figure supplement(s) for figure 1:

**Source data 1.** xlsx.

**Figure supplement 1.** SOX21 is unilaterally detected in the developing trachea and overexpression of SOX21 leads to lung cysts.

(*Rawlins et al., 2009a*; *Figure 1A*). SOX21 was never observed in the distal buds, but always in cells that also express SOX2. SOX21 shows different expression levels among cells during the early developmental stages (E12.5–E14.5), in stark contrast to the more homogeneous distribution of SOX2 (*Figure 1A*). At E18.5, SOX21 is homogeneously expressed in the trachea, heterogeneously expressed in the main bronchi and absent in the intrapulmonary airways (*Figure 1A*). In summary, SOX21 is expressed specifically in the extrapulmonary airways (trachea and main bronchi) during the formation of the airway tree, which is the most proximal part of the SOX2+ epithelium.

We also observed a unilateral expression of SOX21 in the main bronchi, starting from the bifurcation of the trachea (the carina) (*Figure 1A*, *Figure 1—figure supplement 1A*). SOX21 is abundantly expressed on the medial side of the airway, but almost completely absent on the lateral side (*Figure 1—figure supplement 1A*). This unilateral expression pattern is present during branching morphogenesis but becomes less apparent after E15.5 (*Figure 1—figure supplement 1A*). Previously, SOX2 was found to be higher expressed in the dorsal compared to the ventral side of the tracheal epithelium (TE) during development (*Que et al., 2009*). In the developing trachea, the ventral side is marked by SOX9+ cartilage nodules and the dorsal side by smooth muscle actin+ (SMA+) mesenchymal cells (*Hines et al., 2013*). We therefore made transverse sections from the trachea up to the main bronchi and observed a more abundant expression of SOX21 on the dorsal side, comparable to the location of high SOX2 levels (*Figure 1B*). We hypothesize that the high expression of SOX2 is important to set up the expression pattern of SOX21 during lung development.

To validate whether high SOX2 levels can induce *Sox21* expression, we ectopically expressed a MYC-tagged SOX2 or -SOX21 in peripheral epithelial lung cells using a doxycycline-inducible transgene and a surfactant protein C (SPC) promoter-driven rtTA (iSox2$^{SPC-rtTA}$ or iSox21$^{SPC-rtTA}$) (*Perl et al., 2002*; *Gontan et al., 2008*; *Figure 1—figure supplement 1B*). Both MYC-SOX2 and MYC-SOX21 expression caused the appearance of cystic structures, albeit much smaller in the MYC-SOX21 model (*Figure 1—figure supplement 1C*). Although a similar phenotype was observed, MYC-SOX2-expressing cells in the distal lung buds lost SOX9 while inducing *Sox21*, but MYC-SOX21-expressing cells in the distal bud retained SOX9+ and did not induce *Sox2* expression. In the airway, MYC-SOX21+ cells were SOX2 positive, although some cells still expressed SOX9, suggesting that these cells are transitioning from a distal to a proximal cell fate (*Figure 1C*). We therefore conclude that SOX2 alone can set up *Sox21* expression, but SOX21 alone is not sufficient to induce a SOX2+ airway cell fate in the in the presence of distal mesenchymal signaling.

## SOX21 and SOX2 are co-expressed in the zone where progenitor cells differentiate

SOX21 marks a specific proximal region of the airway tree during development, and we therefore asked the question what the role of SOX21 is in this distinct zone 1 versus zone 2, which only contains SOX2+ cells. Early during development, at E11.5, SOX2 and SOX21 were both expressed in the lung, esophagus, and thymus. At this embryonic age, the esophagus and thymus already contained TRP63+ epithelial cell progenitors (*Figure 2—figure supplement 1A*), while TRP63+ basal cells only start to appear in the lung at E12.5 (*Figure 2A*). Previously, we showed that SOX2 directly regulates *Trp63* expression (*Ochieng et al., 2014*), and we therefore analyzed whether SOX21 plays a role in the differentiation of airway progenitor cells to basal cells. We found that TRP63+ basal cells appear in zone 1 from E12.5 onward, but not in zone 2 (*Figure 2A*). At E14.5, we found higher

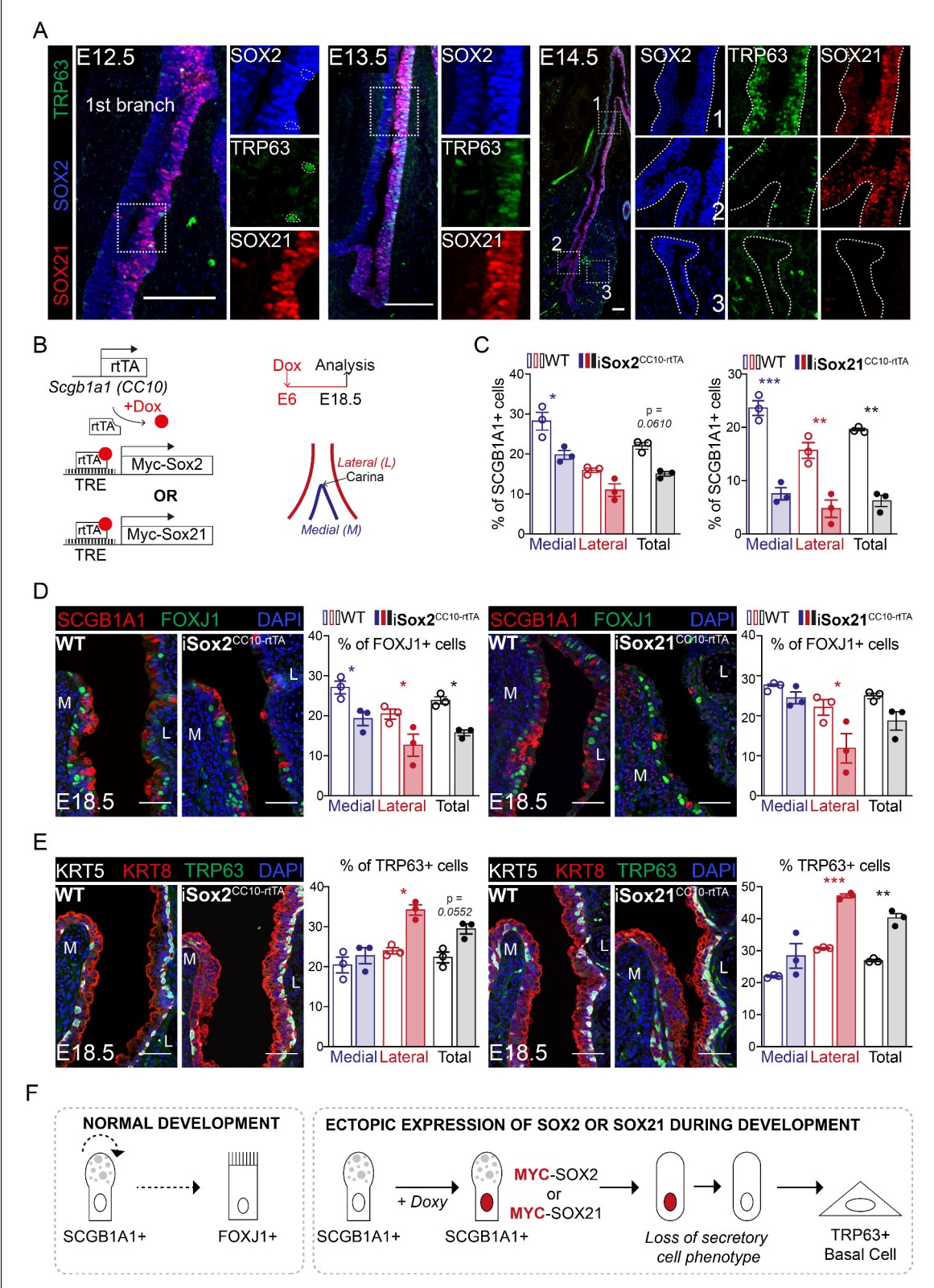

**Figure 2.** SOX2 and SOX21 are co-expressed in the zone where progenitor cells differentiate to basal cells. (**A**) Immunofluorescence of SOX2 (blue), SOX21 (red), and TRP63 (green) on lung sections of E12.5, E13.5, and E14.5 showing a spatial co-localization of the three proteins. Scale bar = 100 μm. (**B**) Schematic representation of the iSox2^CC10-rtTA and iSox21^CC10-rtTA mouse models. Doxycycline was given to the dams from E6 till E18.5. The percentage of secretory, ciliated, and basal cells was analyzed at the medial and lateral side in a 400 μm area around the carina. (**C**) Quantification of

*Figure 2 continued on next page*

*Figure 2 continued*

the percentage of SCGB1A1+ secretory cells at E18.5 in wild-type (WT; open bars), iSox2$^{CC10-rtTA}$, and iSox21$^{CC10-rtTA}$ mice (closed bars) represented as percentage of medial (blue), lateral (red), and total (grey) cells. Two-way ANOVA (n = 3; *p<0.05, **p<0.01, ***p<0.001). Scale bar = 50 µm. (D) Immunofluorescence of SCGB1A1 (red) and FOXJ1 (green) in wild-type (WT), iSox2$^{CC10-rtTA}$, and iSox21$^{CC10-rtTA}$ lung sections at E18.5. Quantification of the percentage of FOXJ1+-ciliated cells at E18.5 in wild-type (WT; open bars), iSox2$^{CC10-rtTA}$, and iSox21$^{CC10-rtTA}$ mice (closed bars) represented as percentage of medial (blue), lateral (red), and total (grey) cells. Two-way ANOVA (n = 3; *p<0.05). Scale bar = 50 µm. (E) Immunofluorescence of TRP63 (red), KRT5 (grey) basal cells, and KRT8 (red) luminal cells in wild-type (WT), iSox2$^{CC10-rtTA}$, and iSox21$^{CC10-rtTA}$ lung sections at E18.5. Quantification of the percentage of TRP63+ basal cells at E18.5 in wild-type (WT; open bars), iSox2$^{CC10-rtTA}$, and iSox21$^{CC10-rtTA}$ (closed bars) mice represented as percentage of medial (blue), lateral (red), and total (grey) cells. Two-way ANOVA (n = 3; *p<0.05, **p<0.01, ***p<0.001). Scale bar = 50 µm. (F) In normal lung development, secretory cells can renew or give rise to ciliated cells. Upon induction of MYC-SOX2 or MYC-SOX21, the secretory cell identity is rapidly lost as shown by the loss of SCGB1A1 expression. Concomitantly, the SCGB1A1/CC10-driven MYC transgene is no longer expressed and cells adopt a TRP63+ basal cell fate.

The online version of this article includes the following source data and figure supplement(s) for figure 2:

**Source data 1.** xlsx.
**Figure supplement 1.** Basal cells arise in the SOX2+ SOX21+ region.

expression of SOX2 and SOX21 as well as more basal cells in the epithelium in close proximity to SMA+ mesenchymal cells (*Figure 1*, *Figure 2—figure supplement 1B,C*). This localization of basal cells is consistent with previous findings (*Que et al., 2009*). The specific localization suggests that mesenchymal–epithelial crosstalk is involved in the combined SOX21 and SOX2 expression and in the progenitor to basal cell differentiation (*Figure 2—figure supplement 1D*).

In our iSox2$^{SPC-rtTA}$ and iSox21$^{SPC-rtTA}$ mouse models, zone 1 is extended to the distal buds and the airways lack zone 2, where only SOX2 is expressed (*Figure 2—figure supplement 1E*). Within the extended SOX2+SOX21+ zone 1, TPR63+ basal cells were also present, showing that induction of both SOX2 and SOX21 is sufficient for the differentiation of airway progenitors to TPR63+ basal cells and can be independent on anatomical localization (*Figure 2—figure supplement 1E*). Previously, conditional deletion of β-catenin in SPC+ cells during lung development caused the appearance of cystic structures, just like after SOX2 and SOX21 induction. Upon β-catenin deletion, an increased number of TRP63+ basal cells was found, of which a subset co-expressed SOX9 (*Ustiyan et al., 2016*). Similarly, we found SOX9+ basal cells in the iSox2$^{SPC-rtTA}$ and iSox21$^{SPC-rtTA}$ mouse lungs, while only sporadic SOX9+ basal cells were detected in the control (*Figure 2—figure supplement 1E*).

To further investigate the effect of SOX2 and SOX21 in proximal differentiated airway epithelium during development, we induced expression of MYC-tagged *Sox2* or *Sox21* transgenes in secretory cells using a CC10/SCGB1A1-driven rtTA-inducible model (iSox2$^{CC10-rtTA}$ and iSox21$^{CC10-rtTA}$) (*Tichelaar et al., 2000*). Doxycycline was giving from E6 onwards to target as much secretory cells as possible during airway development (*Figure 2B*). The number of secretory, basal, and ciliated cells were counted on the medial and lateral side of the main bronchi, immediate distal of the carina (*Figure 2B*). In both models, we found a decreased number of secretory cells (*Figure 2C*), a slight decrease of total number of ciliated cells (*Figure 2D*, grey bar) and an increase in the total number of basal cells (*Figure 2E*, grey bar). Reminiscent of the unilateral expression of SOX21 during development, TRP63+ basal cell numbers increased mostly on the lateral side of the airway, while a decrease in secretory cells was observed on both sides. This suggests that a difference in medial and lateral mesenchymal-epithelial signaling directs the induced differentiation of MYC-SOX2 or MYC-SOX21 expressing cells. Focusing on the MYC expressing cells, we observed that these cells were mostly negative for basal cell marker TRP63, secretory cell marker SCGB1A1, and ciliated cell marker FOXJ1 (*Figure 2—figure supplement 1F*). This suggests that upon the induction of MYC-SOX2 or MYC-SOX21, the secretory cell identity is rapidly lost as shown by the loss of SCGB1A1 expression. Concomitantly, the SCGB1A1/CC10-driven MYC transgene is no longer expressed and cells adopt a TRP63+ basal cell fate (*Figure 2F*).

## SOX21 and SOX2 regulate maintenance and differentiation of the airway progenitor state

Next, we studied how reduced levels of SOX21 affect the differentiation of progenitor cells during development in *Sox21* heterozygous (*Sox21*$^{+/−}$) and homozygous (*Sox21*$^{−/−}$) knockout mice.

$Sox21^{-/-}$ mice do not show respiratory distress at birth (**Kiso et al., 2009**). Their lungs are smaller compared to wild-type littermate controls but have no apparent branching defect (data not shown). To analyze how reduced levels of SOX21 affect epithelial differentiation, we evaluated airway specific cells numbers at E14.5 and E18.5 distal to the *carina*, at the medial side where SOX2 and SOX21 levels are highest (**Figure 1—figure supplement 1A**). At E14.5, we found an increased number of basal cells in $Sox21^{+/-}$ mice, which was even more pronounced in $Sox21^{-/-}$ mice (**Figure 3A**). In contrast, a decrease in the number of basal cells is observed in lungs of $Sox2^{+/-}$ at E14.5 (**Figure 3—figure supplement 1A**), corresponding with the dose-dependent role described for SOX2 in airway differentiation (**Que et al., 2007**). Hence, reduced levels of SOX2 and SOX21 appear to oppositely affect airway progenitor cell differentiation. SOX21 maintains the SOX2 progenitor state, while SOX2 expression is needed to initiate progenitor to basal cell differentiation.

To determine the role of SOX2 and SOX21 for differentiation to airway-specific cell types, we investigated the differentiation to ciliated cells. Ciliated cells started to appear at E14.5 and can be quantified using TRP73, an early marker for differentiation, and FOXJ1, a more mature marker (**Marshall et al., 2016**). At E14.5, there was a small increase of TRP73+- and FOXJ1+-ciliated cells in $Sox21^{+/-}$ airways and an even larger increase in $Sox21^{-/-}$ airways (**Figure 3B,C**), reaffirming that SOX21 levels are important for maintenance of the progenitor state and suppression of differentiation. In the $Sox2^{+/-}$ mice, TRP73+- and FOXJ1+-ciliated cell numbers did not differ from wild-type controls (**Figure 3—figure supplement 1B,C**), suggesting that decreasing SOX2 levels do not influence the initiation of ciliated cell differentiation. Moreover, overall proliferation was unaltered during development in $Sox2^{+/-}$ airways and only slightly increased in $Sox21^{-/-}$ airways (**Figure 3D**, **Figure 3—figure supplement 1D**), suggesting that SOX2 and SOX21 act mainly on differentiation and not proliferation.

At E18.5, the increase in basal cell numbers in $Sox21^{-/-}$ observed at E14.5 was no longer detectable, although increased numbers of ciliated cells were still present. Additionally, we also observed a decrease in secretory cells at E18.5 (**Figure 3E**). Thus, it seems that in the absence of SOX21 the increase in ciliated cells is at the expense of secretory cells, in agreement with the reported finding that the secretory cell lineage gives rise to ciliated cells (**Rawlins et al., 2009b**). In $Sox2^{+/-}$ lungs, only a minor decrease in secretory cells was observed at E18.5 (**Figure 3—figure supplement 1E**).

To better understand the effect of SOX21 on SOX2-induced gene expression, we analyzed promotor regions that are activated upon expression of SOX2 by luciferase assays. We found that SOX2 activates its own promoter, as well as the promoter regions of the *Trp63* and *Trp73* genes (**Figure 3F**, **Figure 3—figure supplement 1F**). SOX21 binding motifs have been characterized through chromatin immunoprecipitation (**Matsuda et al., 2012**), and we subsequently identified putative binding sites for SOX21 in the promoter regions of *Sox2*, *Trp63*, and *Trp73* (**Figure 3F**, **Figure 3—figure supplement 1F**). When increasing amounts of SOX21 were titrated in with SOX2, luciferase activity with the *Trp63* promotor significantly decreased (**Figure 3F**, **Figure 3—figure supplement 1F**). This shows that SOX21 can suppress promotor regions of SOX2 target genes.

Taken together, we propose that high levels of SOX2 initiate SOX21 expression, marking a zone of the proximal airways where progenitor cells start to differentiate. Within this zone, SOX21 and SOX2 in progenitor cells have a regulatory function, by maintaining a balance between progenitor maintenance and differentiation (**Figure 3G**).

## SOX2 drives and SOX21 represses basal cell differentiation to ciliated cells

At E18.5, SOX2 and SOX21 remain expressed in the extrapulmonary airways, the region where basal cells are maintained. Basal cells are important for regeneration after injury, as they can proliferate and differentiate to secretory and ciliated cells (**Rock et al., 2009**). To better understand how SOX2 and SOX21 regulate the differentiation of basal cells, we used an in vitro differentiated ALI culture method of mouse tracheal epithelial cells (MTECs; **Eenjes et al., 2018**). ALI cultures provide standardized conditions to study basal cell differentiation at different time points. Using this model, we show that *Sox2* and *Sox21* were both expressed at baseline levels at the start of ALI and that their levels gradually increased in the initial days of ALI differentiation (**Figure 4A**). Genes reflecting the differentiation status of ciliated and secretory cells were increasingly expressed in a similar fashion (**Figure 4—figure supplement 1A**). Immunofluorescence analysis on ALI day 10 showed expression of SOX2 and SOX21 in basal and apical cells, although both proteins were heterogeneously

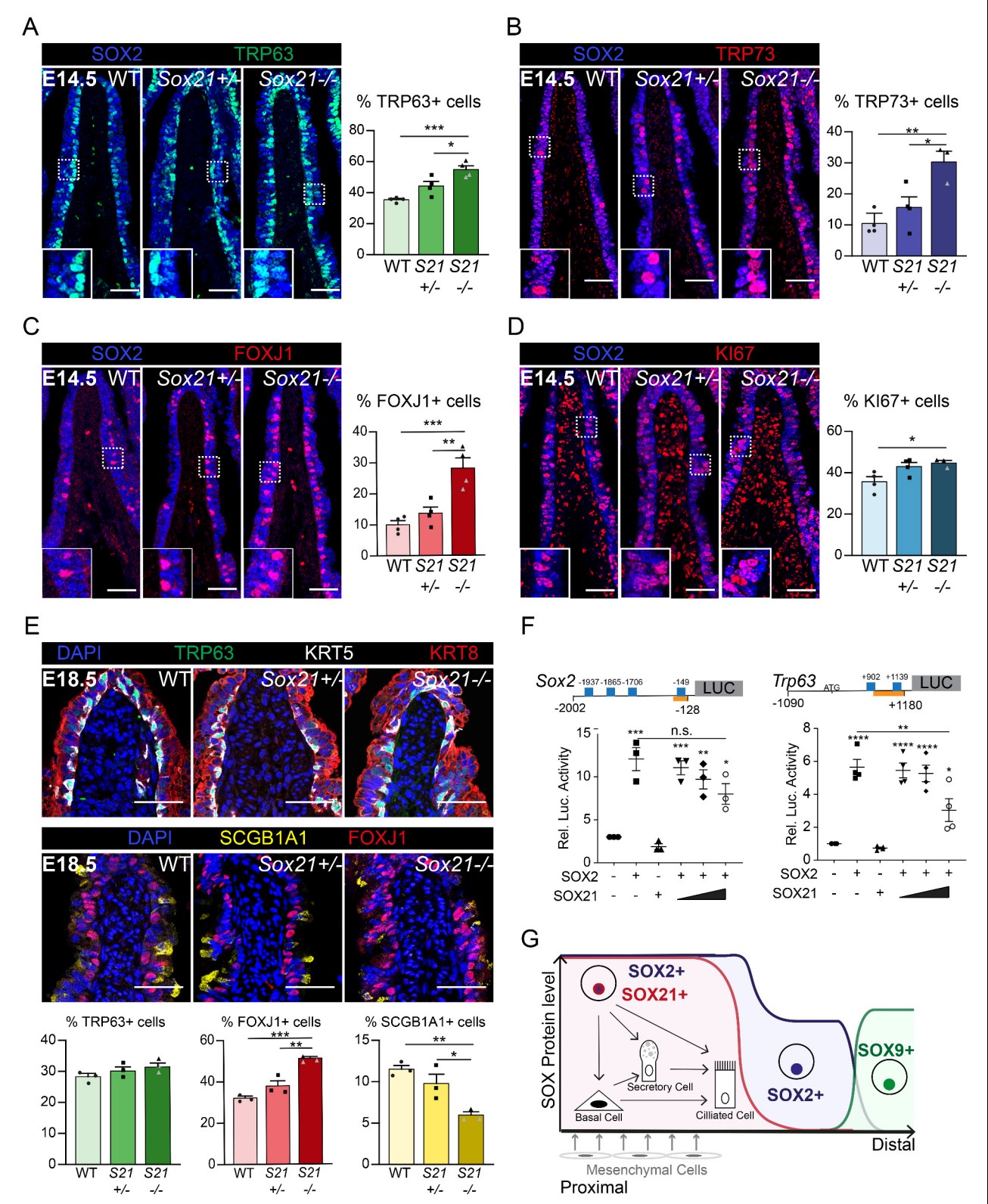

**Figure 3.** SOX21 counter balances SOX2+ progenitor differentiation to airway-specific cell types. (A–D) Immunofluorescence and quantification of the number of TRP63+ basal cells (A), TRP73+ cells (B), FOXJ1+-ciliated cells (C), and KI67+-dividing cells (D) at E14.5 in wild-type (WT), *Sox21⁺/⁻*, and *Sox21⁻/⁻* mice. The number of cells were counted within the first 400 μm immediately distal of the carina at the medial side of the airway. One-way ANOVA (n = 3; *p<0.05). Scale bar = 50 μm. (E) Immunofluorescence and quantification of the number of TRP63+ (green) basal cells, FOXJ1+ (red)

*Figure 3 continued on next page*

*Figure 3 continued*

ciliated cells, and SCGB1A1+ (yellow) secretory cells at E18.5 in wild-type (WT), *Sox21*[+/−], and *Sox21*[−/−] mice. The number of cells were counted within the first 400 μm immediately distal of the carina at the medial side of the airway. One-way ANOVA (n = 3; *p<0.05, **p<0.01, ***p<0.001). Scale bar = 50 μm. (F) Luciferase assay to test the transcriptional activity of the *Sox2* promotor region from −2002 till −128 and *Trp63* promotor region from −1090 till +1180 (+1 is considered the transcriptional start site). Blue squares showing SOX21 binding motifs and the orange bar shows the region bound by RNA polymerase II. The graph shows luciferase activity induced after transfection of FLAG-SOX2 and/or increasing amounts of MYC-SOX21. One-way ANOVA (n = 3; **p<0.01, ***p<0.001, ****p<0.0001, when not indicated otherwise, stars show significance compared to control). (G) Schematic overview of epithelial progenitor maintenance and differentiation during murine lung development. From proximal to distal a transition of SOX protein levels takes place. The most proximal part of the lung, the trachea, and the two main bronchial branches, show high expression levels of SOX2 and co-expression of SOX21. SOX21 balances the maintenance of the SOX2 progenitor state in a zone where progenitor cells are prone to differentiate to airway-specific cell types.

The online version of this article includes the following source data and figure supplement(s) for figure 3:

**Source data 1.** xlsx.

**Figure supplement 1.** SOX21 counter balances SOX2+ progenitor differentiation to airway specific cell types.

distributed among the two cell layers (*Figure 4B*). To determine whether SOX2 and SOX21 levels correlated, we measured the fluorescence intensity of both proteins (*Figure 4—figure supplement 1B*). Cells highly expressing SOX2 were frequently high in SOX21 expression and vice versa (*Figure 4—figure supplement 1C*). Basal cells were mainly Sox2[low]Sox21[low] (80.5%), while ciliated cells were either SOX2[high]SOX21[high] (77.7%) or SOX2[high]SOX21[low] (14.5%; *Figure 4C*). In the adult mouse TE, SOX2 and SOX21 levels were similar between cell types in vivo (*Figure 5A*), suggesting that these are stable for the maintenance of TE and only increase when differentiation is initiated (*Figure 4—figure supplement 1D*).

To test the importance of SOX2 and SOX21 levels in basal cell differentiation, we analyzed ALI day 10 cultures of WT, *Sox2*[+/−], and *Sox21*[+/−] mice. After 10 days, there were comparable numbers of basal cells in *Sox2*[+/−], *Sox21*[+/−], and WT ALI cultures (*Figure 4—figure supplement 1C*). In *Sox21*[+/−] ALI cultures, more basal cells differentiated to FOXJ1+ ciliated cells compared to WT (*Figure 4D*). TRP73 is one of the earliest markers expressed by basal cells upon initiation of ciliated cell differentiation, followed by the loss of TRP63 expression (*Marshall et al., 2016*). While there was an increase in FOXJ1+ ciliated cells, TRP73+/TRP63+ or single TRP73+ cell numbers remained stable (*Figure 4—figure supplement 1C*). In *Sox2*[+/−] ALI cultures, fewer basal cells differentiated to ciliated cells compared to WT, as was shown by a significant decrease of both single TRP73+ and FOXJ1+ cells (*Figure 4D*; *Figure 4—figure supplement 1C*). The differentiation capacity towards secretory cells was not affected in either *Sox2*[+/−] or *Sox21*[+/−] ALI cultures compared to WT (*Figure 4D*). Therefore, SOX2 and SOX21 levels are mainly involved in balancing the differentiation of basal cells to ciliated cells, with SOX2 stimulating differentiation and SOX21 inhibiting differentiation (*Figure 4—figure supplement 1D*).

To further understand the role of SOX2 and SOX21 in differentiated airway epithelium, we used iSox2[CC10-rtTA] and iSox21[CC10-rtTA] adult mice to induce expression of *Sox2* or *Sox21* in secretory cells for 6 weeks (*Figure 4E*). Similar to the CC10-rtTA induction of MYC-SOX2 or MYC-SOX21 during development, the MYC transgene expressing cells were mostly negative for other cell-specific markers (*Figure 4—figure supplement 1E*). Induction of either MYC-SOX2 or MYC-SOX21 resulted in a decrease in secretory cells, while MYC-SOX2 also increased TRP63+ basal cell numbers (*Figure 4E*). Thus, MYC-SOX2 induction in secretory cells drives the appearance of basal cells. In addition, the induction of MYC-SOX21 in secretory cells of adult airway epithelium resulted in a loss of secretory phenotype but appears insufficient to drive differentiation toward a basal or ciliated cell type (*Figure 4F*).

## Deficiency of SOX2 decreases airway epithelial repair after naphthalene-induced injury while SOX21 stimulates it

Because fine-tuning of SOX2 and SOX21 levels is important in basal cell differentiation in vitro, we investigated whether SOX21 is also important for basal cell differentiation after injury in vivo. SOX21 remains expressed throughout the tracheal epithelium (TE) in both basal and non-basal cells in adult mice (*Figure 5A*). In addition, we observed SOX21-expressing cells in the submucosal glands (SMG), which also contribute to TE regeneration after injury (*Lynch et al., 2018*; *Tata et al., 2018*;

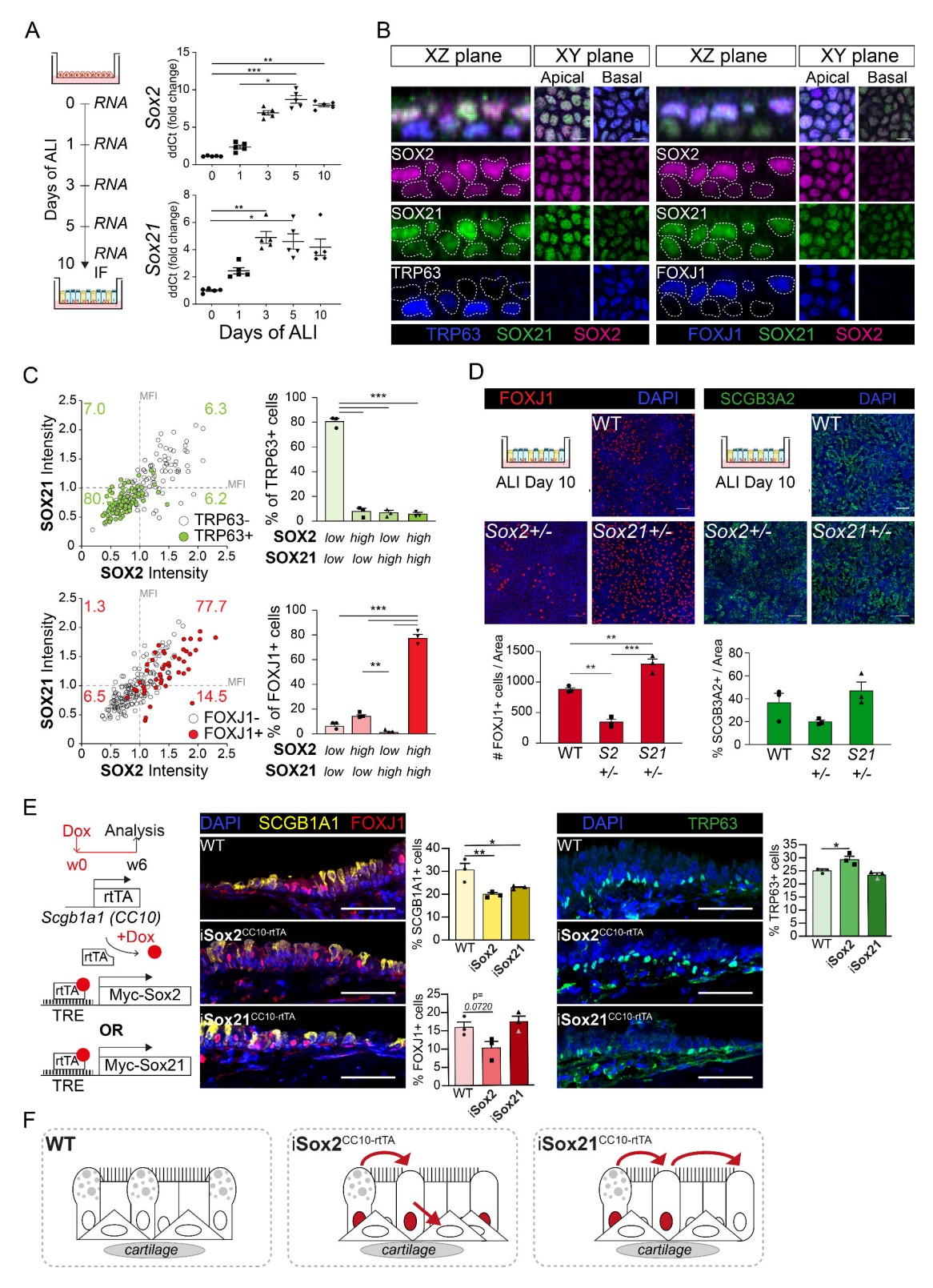

**Figure 4.** SOX2 and SOX21 are inversely correlated with basal cell differentiation to ciliated cells. (**A**) Schematic overview of mouse tracheal epithelial cell (MTEC) culture. QPCR analysis of *Sox2* and *Sox21* expression during differentiation of MTECs on air–liquid interface (ALI). One-way ANOVA (n = 5; *p<0.05, ** p<0.01, ***p<0.001). (**B**) Immunofluorescence staining on MTEC 10 days after ALI of SOX2 (purple), SOX21 (green), and TRP63 (blue, left images) or FOXJ1 (blue, right images). Dotted lines show representation of measured nuclei for fluorescence intensity. Scale bar = 25 µm. (**C**) Dot-plot

*Figure 4 continued on next page*

Figure 4 continued

of the MFI of SOX2 (x-axis) and SOX21 (y-axis). The green filled circles are TRP63+ basal cells and the number in each quadrant represents the percentage of basal cells in each quadrant. Bar graph is the quantification of basal cells that are either high (MFI > 1) or low (MFI < 1) in expression of SOX2 and SOX21. The red filled circles are FOXJ1+-ciliated cells and the number in each square represents the percentage of ciliated cells in each quadrant. Bar graph is the quantification of ciliated cells that are either high (MFI > 1) or low (MFI < 1) in expression of SOX2 and SOX21. One-way ANOVA (n = 3; **p<0.01, ***p<0.001). (D) Analysis of MTEC cultures of tracheal cells derived from wild-type (WT), $Sox2^{+/-}$, or $Sox21^{+/-}$ animals after 10 days of ALI culture. Immunofluorescence staining of ciliated cells (FOXJ1, red) or secretory cells (SCGB3A2, green). Scale bar = 50 µm. Quantification of the number of ciliated cells per 775 µm² field. One-way ANOVA (n = 3; **p<0.01). (E) Schematic representation of the iSox2$^{CC10-rtTA}$ and iSox21$^{CC10-rtTA}$ mouse models. Doxycycline was given to adult mice for 6 weeks. Immunofluorescence and quantification of the number of SCGB1A1+ (yellow) secretory cells, FOXJ1+ (red) ciliated cells and TRP63+ (green) basal cells in wild-type (WT) compared to iSox2$^{CC10-rtTA}$ and iSox21$^{CC10-rtTA}$ mice. One-way ANOVA (n = 3; *p<0.05, **p<0.01). Scale bar = 50 µm. (F) Upon homeostasis, tracheal epithelium has a low turnover. The induction of MYC-SOX2 or MYC-SOX21 results in a loss of secretory cell phenotype. The induction of MYC-SOX2 drives the appearance of basal cells. The induction of MYC-SOX21 appears insufficient to drive differentiation toward a basal or ciliated cell type.

The online version of this article includes the following source data and figure supplement(s) for figure 4:

Source data 1. xlsx.

Figure supplement 1. SOX2 and SOX21 are inversely correlated with basal cell differentiation to ciliated cells.

Figure 5A). To test whether SOX21 and SOX2 levels are important to control adult stem cell differentiation, we exposed wild-type (WT), $Sox2^{+/-}$, and $Sox21^{+/-}$ mice to corn oil (CO) or naphthalene to induce transient epithelial injury. We examined the immediate response after 2 days and the recovery after 5 and 20 days post-injury (DPI) (Figure 5B). Due to the fragility of $Sox21^{-/-}$ mice, we were unable to study adult $Sox21^{-/-}$ TE after naphthalene injury (Kiso et al., 2009). Neither $Sox2^{+/-}$ nor $Sox21^{+/-}$ TE showed significant differences in basal, dividing basal, or ciliated cell numbers compared to WT after CO exposure (Figure 5—figure supplement 1D).

SOX9+ SMG cells surface at the TE after administering a high naphthalene dose and thereby contribute to repair of the TE (Lynch et al., 2018; Tata et al., 2018). The high concentration of naphthalene administered to the mice causes severe injury by removing most luminal epithelial cells and leaving few basal cells (Figure 5—figure supplement 1A), as reported (Cole et al., 2010; Tata et al., 2018). Previously, it was shown that SOX2 expression was largely extinguished from SOX9+ SMG cells at 1 DPI, suggesting that changes in SOX2 expression are important in the early contribution of SMG cells to repair (Lynch et al., 2018). However, we found that SOX2 or SOX21 expression was unaltered at 2 DPI compared to CO exposed mice, suggesting that SOX2 downregulation at 1 DPI is temporary and a consequence of naphthalene administration (Figure 5A, Figure 5—figure supplement 1B). After 5 days, we observed a decrease of SOX9+ cells in $Sox2^{+/-}$ TE, and a small increase of SOX9+ cells in $Sox21^{+/-}$ TE compared to WT and to each other (Figure 5C). This shows that SOX2 and SOX21 levels are important for TE regeneration by SOX9+ SMG cells.

To determine whether SOX2 or SOX21 deficiency affects regeneration after naphthalene injury, we quantified the percentage of secretory, ciliated, non-dividing, and dividing basal cells. The number of ciliated and dividing basal cells was similar between WT, $Sox21^{+/-}$, and $Sox2^{+/-}$ at 5 DPI (Figure 5—figure supplement 1C). At 20 DPI, fewer FOXJ1+ ciliated cells, but more dividing and non-dividing basal cells were observed in $Sox2^{+/-}$ TE compared to WT, $Sox21^{+/-}$, or to $Sox2^{+/-}$ CO exposed mice (Figure 5D–E, Figure 5—figure supplement 1D). Both WT and $Sox21^{+/-}$ TE had a similar number of ciliated cells, non-dividing basal cells, and dividing basal cells at 20 DPI compared to CO exposed mice (Figure 4D, Figure 5—figure supplement 1D). Thus, recovery of injury in the WT and $Sox21^{+/-}$ TE appears the same, while $Sox2^{+/-}$ TE show a delayed recovery (Figure 5E, Figure 5—figure supplement 1D), indicating an important role for SOX2 in TE regeneration.

## SOX2 and SOX21 expression is conserved in human airway epithelial cells

The data above show that SOX21 is an important factor to maintain the balance between progenitor maintenance and differentiation in both mouse lung development and adult basal cell differentiation. Next, we asked whether this role is conserved in human airway epithelial development. SOX21 is enriched in SOX2+ stalk (differentiating bronchiole) versus tip (distal) progenitor cells of the human fetal lung (Nikolić et al., 2017). We observed sporadic SOX21-expressing cells in lung sections at post-conceptional week (PCW) 17. The SOX21-expressing cells were mainly found in larger

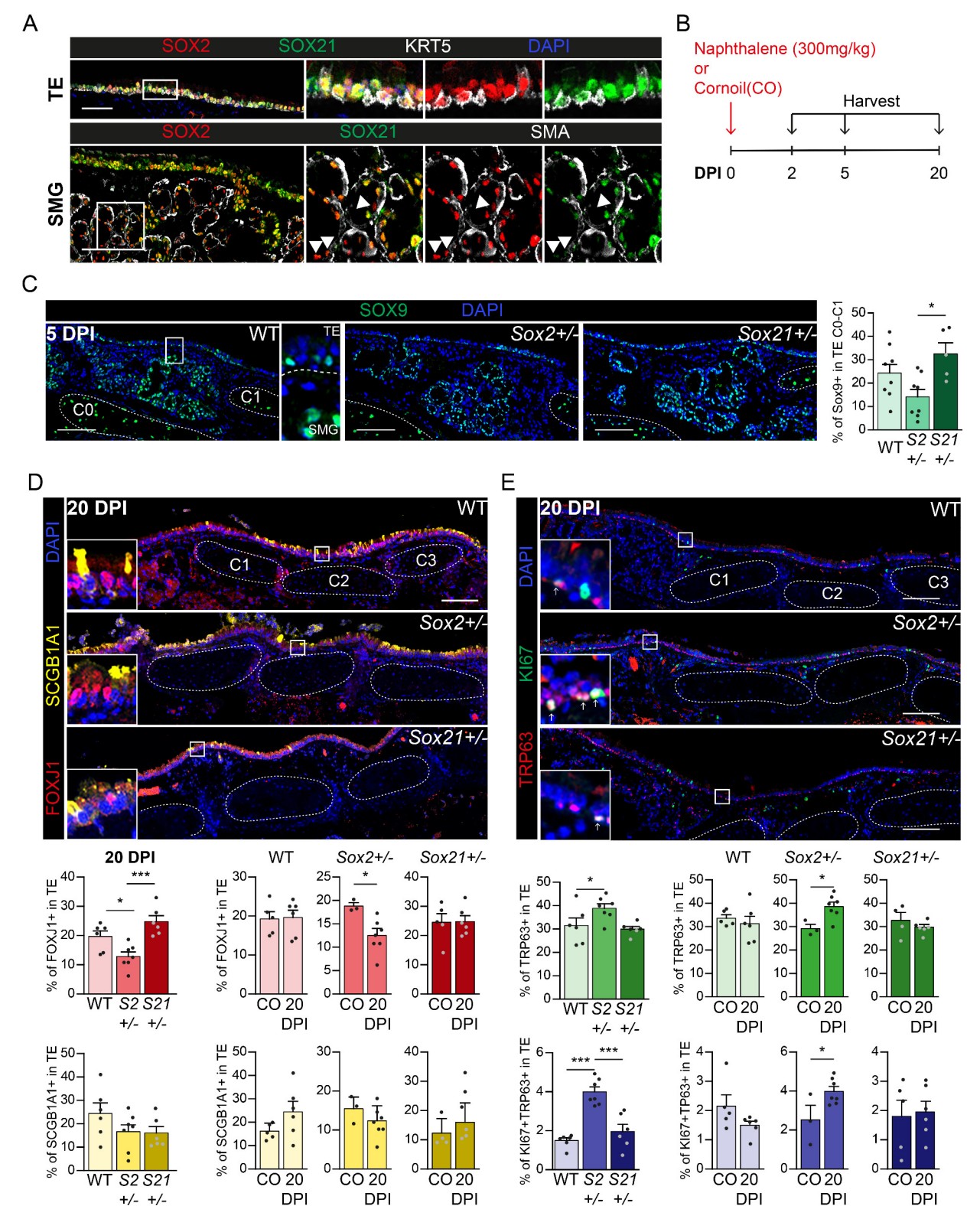

**Figure 5.** Regeneration is delayed in SOX2 deficient tracheal epithelium. (**A**) Immunofluorescence staining on tracheal sections of adult wild-type mice for SOX2 (red), SOX21 (green), and KRT5 (grey, top row) or smooth muscle actin (grey, SMA, bottom row). TE = tracheal epithelium. SMG = submucosal gland. Closed arrowheads (u) indicate single SOX2+ cells. Scale bar = 100 μm. (**B**) Schematic overview of the experimental set up of the Naphthalene injury and recovery in wild-type (WT), *Sox2^{+/−}* (*S2^{+/−}*), and *Sox21^{+/−}* (*S21^{+/−}*) mice. (**C**) Immunofluorescence and quantification of the number of SOX9+

*Figure 5 continued on next page*

*Figure 5 continued*

cells, 5 days post injury (DPI), in the upper TE from Cartilage (C) ring 0 till C1 in WT, *Sox2*$^{+/-}$ and *Sox21*$^{+/-}$ mice. Scale bar = 100 µm. One-way ANOVA (WT n = 8, *Sox2*$^{+/-}$ n = 8, *Sox21*$^{+/-}$ n = 5, *p<0.05). (D) Immunofluorescence of FOXJ+ (red) cells and SCGB1A1+ (yellow) cells in the TE, of WT, *Sox2*$^{+/-}$ and *Sox21*$^{+/-}$ mice at 20 DPI. Scale bar = 100 µm. Quantification of the number of FOXJ1+-ciliated cells and SCGB1A1+ secretory cells from C0 till C6 between genotypes. One-way ANOVA (WT n = 6, *Sox2*$^{+/-}$ n = 7, *Sox21*$^{+/-}$ n = 6, *p<0.05, ***p<0.001). Quantification of the number of ciliated cells, 20 DPI compared to the corn oil (CO) exposed mice of each WT, *Sox2*$^{+/-}$ and *Sox21*$^{+/-}$ mice. Unpaired T-test (WT: CO n = 5 and 20DPI n = 6, *Sox2*$^{+/-}$: CO n = 3 and 20DPI n = 7, *Sox21*$^{+/-}$: CO n = 5 and 20DPI n = 6, *p<0.05). (E) Immunofluorescence with TRP63 (red) and KI67 (green) to mark dividing basal cells in the TE, 20 DPI, in WT, *Sox2*$^{+/-}$, and *Sox21*$^{+/-}$ mice. Scale bar = 100 µm. Quantification of the number of dividing basal cells from C0 till C6 between genotypes. One-way ANOVA (WT n = 6, *Sox2*$^{+/-}$ n = 7, *Sox21*$^{+/-}$ n = 6, ***p<0.001). Quantification of dividing basal cells, 20 DPI compared to CO mice of each wild-type (WT), *Sox2*$^{+/-}$, and *Sox21*$^{+/-}$. Unpaired T-test (WT: CO n = 5 and 20 DPI n = 6, *Sox2*$^{+/-}$: CO n = 3 and 20 DPI n = 7, *Sox21*$^{+/-}$: CO n = 5 and 20 DPI n = 6, *p<0.05).

The online version of this article includes the following source data and figure supplement(s) for figure 5:

**Source data 1.** xlsx.
**Figure supplement 1.** Regeneration is delayed in SOX2 deficient tracheal epithelium.

airways, where basal cells were also present (*Figure 6A*). Additionally, we cultured human fetal lung tip (PCW 17) organoids as previously described (*Figure 7—figure supplement 1A*; *Nikolić et al., 2017*). All cells within the organoids were positive for SOX2 and co-expressed either SOX9, SOX21, or both, unlike human PCW 17 lung sections (*Figure 6A,B*). Some SOX21-expressing cells in the fetal lung organoids were positive for the basal cell marker TP63, just like in human fetal lung sections (*Figure 6A,B*, white arrowheads). To determine whether SOX21 increases upon further lung development, we differentiated human fetal lung tip organoids into a more mature airway organoid. To induce maturation, we replaced fetal lung organoid medium with airway organoid medium as recently described (see Materials and Methods, Table 1; *Sachs et al., 2019*). This caused the lung organoids to take on a differentiated airway epithelium phenotype, with an increase in basal cells and the appearance of ciliated cells (*Figure 6C*). In these airway organoids, SOX21 expression was observed in most cells (*Figure 6C*), and we confirmed expression of SOX21 in adult airway epithelial cells using human bronchial sections (*Figure 6D*).

To functionally assess the role of SOX21 in human adult airway epithelium, we isolated airway epithelial cells from human bronchial tissue. Both SOX2 and SOX21 are expressed throughout the bronchial epithelium (*Figure 6D*). ALI cultures showed that *SOX2* and *SOX21* were also expressed in vitro, as with murine cells. Prolonged culturing resulted in a significant increase of *SOX2* and a slight increase of *SOX21* (*Figure 7A*, *Figure 7—figure supplement 1B*). Both SOX2 and SOX21 were higher expressed in the apical layer compared to the basal layer (*Figure 7B*). Most TP63+ basal cells were low in SOX2 and SOX21 expression (88.3%) (*Figure 7—figure supplement 1D*). Next, we determined whether SOX2$^{high}$ and/or SOX21$^{high}$ cells in the apical layer were FOXJ1+-ciliated cells. Comparing FOXJ1+ and FOXJ1− nuclei showed that ciliated cells were high in SOX2 expression, while SOX21 was highest in non-ciliated cells (*Figure 7B*, *Figure 7—figure supplement 1C*). Fluorescence intensity of SOX2 and SOX21 also showed that most ciliated cells were SOX2$^{high}$SOX21$^{low}$ (75%; *Figure 7—figure supplement 1D*). Furthermore, the intensities of either SOX2 or SOX21 for ciliated (FOXJ1+), basal (TP63+), and FOXJ1− TP63− cells, showed that SOX21 was mainly high in double-negative cells (*Figure 7C*). We think that these double-negative cells are intermediate cells transitioning from basal cells to a ciliated or sercretory cell fate, the para-basal cell. In accordance with this observation, sections of human airway epithelium show SOX21$^{high}$ cells in between the basal and luminal layer, which also express both basal cell marker KRT5 and luminal cell marker KRT8 (*Figure 6D* 1–3, arrows). Comparison of our expression results to published single-cell RNA data on human primary bronchial epithelial ALI culture (*Plasschaert et al., 2018*) confirmed that high levels of *SOX21* mRNA denote an intermediate cell type. Furthermore, in line with our observation, high levels of *SOX2* were observed both in the intermediate state and in differentiated ciliated cells (*Figure 7—figure supplement 1E*).

Finally, we assessed the effects of reduced SOX2 or SOX21 levels on human basal cell differentiation by transducing ALI cultures with lentiviruses expressing short-hairpin RNAs (shRNA) to knockdown *SOX2* or *SOX21*, and a mCherry to mark transduced cells. Cells were transduced at D0 of ALI, and the differentiation to ciliated and MUC5B+ secretory goblet cells was assessed at D14. SCGB1A1+ secretory cells were rarely observed in our cultures and therefore not analyzed. We

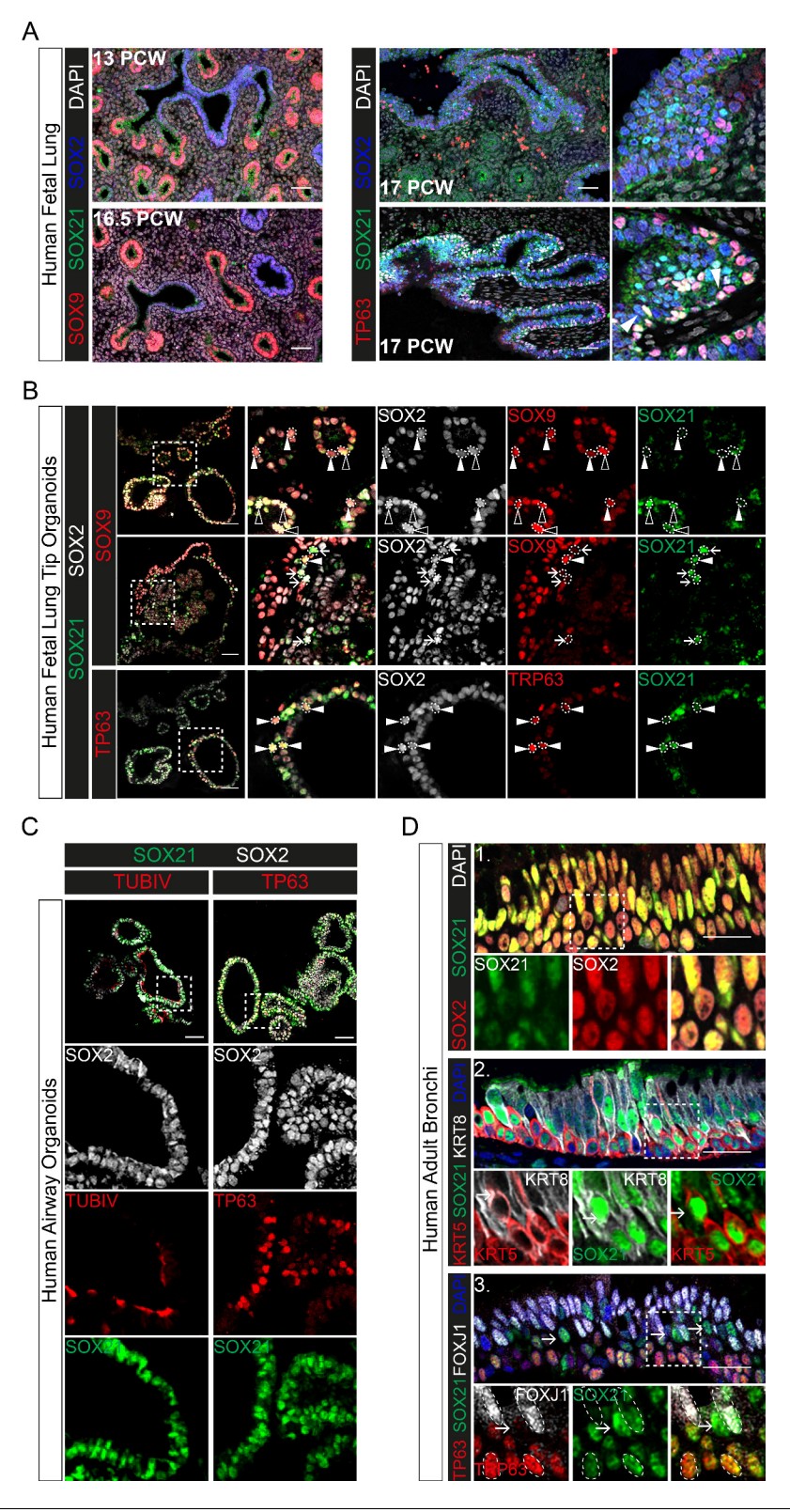

**Figure 6.** Evolutionary conserved expression of SOX21 in human airway epithelium. (**A**) Immunofluorescence on human fetal lung sections post-conceptional week (PCW) 13, 16.5, and 17. Proximal–distal patterning of human developing airways is analyzed with SOX9 (red) and SOX2 (blue) (left). In the small developing airways (left), no expression of SOX21 (green) was observed. On the right, co-staining of SOX2 (blue), SOX21 (green), and TP63 (red). The closed white arrowheads show cells co-expressing TP63 and SOX21. Scale bar = 50 μm. (**B**) Top and middle row: Immunofluorescence

*Figure 6 continued on next page*

*Figure 6 continued*

analysis of fetal lung organoids shows SOX2 (grey), SOX9 (red), and SOX21 (green) positive cells. White closed triangles (▶) show cells positive for SOX2 and SOX9, open triangles (w) show cells positive for SOX2, SOX21, and SOX9. Arrows (g) show cells positive for SOX21 and SOX2. Bottom row: Immunofluorescence analysis of SOX2 (grey), TP63 (red), and SOX21 (green) in fetal lung organoids. White closed arrowheads (▶) show cells positive for SOX2, TP63, and SOX21. Scale bar = 50 μm. (C) Fetal lung organoids differentiated to airway epithelium, as shown by the presence of ciliated cells (TUBIV; left) or basal cells (TP63, right), have an abundant expression of SOX21 (green). Scale bar = 50 μm. (D) Immunofluorescence analysis of sections of human adult bronchi shows co-localization of SOX2 (red) and SOX21 (green) throughout the epithelium (1, top row). SOX21 (green) is expressed in luminal (KRT8; grey) and basal (KRT5; red) cells (2, middle row). SOX21 is expressed in basal (TP63; red) and ciliated (FOXJ1; grey) and is high expressed in cells absent of TP63 and FOXJ1(g) (3, bottom row). Scale bar = 25 μm.

compared the fluorescence intensity of SOX2 or SOX21 in mCherry+ transduced cells with mCherry− non-transduced cells and observed an average lower intensity in two of three shRNAs used (*Figure 7—figure supplement 1F*). We observed an overall increase of FOXJ1+ cells upon knockdown of *SOX2* or *SOX21* in several independent donor ALI cultures (*Figure 7D*). This increase in FOXJ1+ cells was also observed when analyzing the mCherry+ cells only (*Figure 7D*). Thus, knockdown of *SOX2* or *SOX21* in the human airway cells favor a FOXJ1+ cell type, partly resembling the mouse ALI experiments (*Figure 3C*, *Figure 3—figure supplement 1C*). Despite the increase in FOXJ1+ nuclei, we did not observe an increase in fully differentiated ciliated cells as determined by the formation of TubulinIV+ cilia (*Figure 7—figure supplement 1G*). However, the increase in FOXJ1+ nuclei is seemingly combined with an impaired ability to differentiate into MUC5B+ goblet cells (*Figure 7E*). In conclusion, our data suggest that both SOX2 and SOX21 suppress basal to ciliated cell differentiation, while both seem to positively affect goblet cell differentiation in the human airways.

## Discussion

During lung development, a proximal-to-distal epithelial gradient is observed by the separation of proximal SOX2- and distal SOX9-expressing cells. Here we show a further evolutionary conserved regionalization of the proximal epithelium by marking a SOX2+/SOX21+ proximal zone. Within this zone, progenitor cells differentiate into basal cells. With the use of human embryonic lung sections and fetal lung organoids, we confirmed SOX21 expression in SOX2+ progenitor stalk cells (*Nikolić et al., 2017*). In addition, SOX21 becomes more widely expressed when human fetal lung organoids are differentiated into airway organoids. Based on our findings, we propose that (1) SOX2+SOX21+ cells in murine and human fetal lung organoids are progenitor cells in an environment destined for differentiation and (2) that SOX21 is important in balancing maintenance and differentiation of SOX2+ airway progenitor cells.

SOX2 is a key regulator in regulating proliferation and differentiation in many different stem cells populations; however; the chromatin regions targeted by SOX2 are cell-type specific. The regulation of stem cells by SOX2 is dependent on both its co-factors and expression levels (*Brafman et al., 2013*; *Hagey et al., 2018*). We observed a correlation between high SOX2 expression and appearance of SOX21 on the dorsal side of the proximal airways and trachea. Additionally, ectopic induction of SOX2 in distal cells resulted in an upregulation of SOX21. We suggest that SOX2 requires a certain threshold level of expression to induce SOX21. Interestingly, SOX21 expression was first observed during the embryonic stage (E10.5–11.5), at a time when cells adopt a proximally restricted fate for the extrapulmonary airways (*Yang et al., 2018*). We suggest that SOX21 is a downstream effector of SOX2 during formation of the extrapulmonary airways, separating extrapulmonary airway progenitors from intrapulmonary airway which are only SOX2 positive. A similar induction of SOX21 by SOX2 has been described in the four-cell-stage embryo, in embryonic stem cells and in neuronal progenitor cells. The function of SOX21 and its synergy or antagonism with SOX2 activity is highly context dependent (*Kopp et al., 2008*; *Mallanna et al., 2010*; *Chakravarthy et al., 2011*; *Goolam et al., 2016*; *Graham et al., 2003*; *Ohba et al., 2004*; *Sandberg et al., 2005*; *Matsuda et al., 2012*). As an example of the context-dependent function of SOX21, we showed that upon ectopic expression of SOX21, SOX9+ distal progenitor cells were maintained, with no differentiation to SOX2+ progenitor cells. Thus, SOX21 itself is not capable of driving differentiation to SOX2+ progenitor cells in the presence of distal mesenchymal signaling.

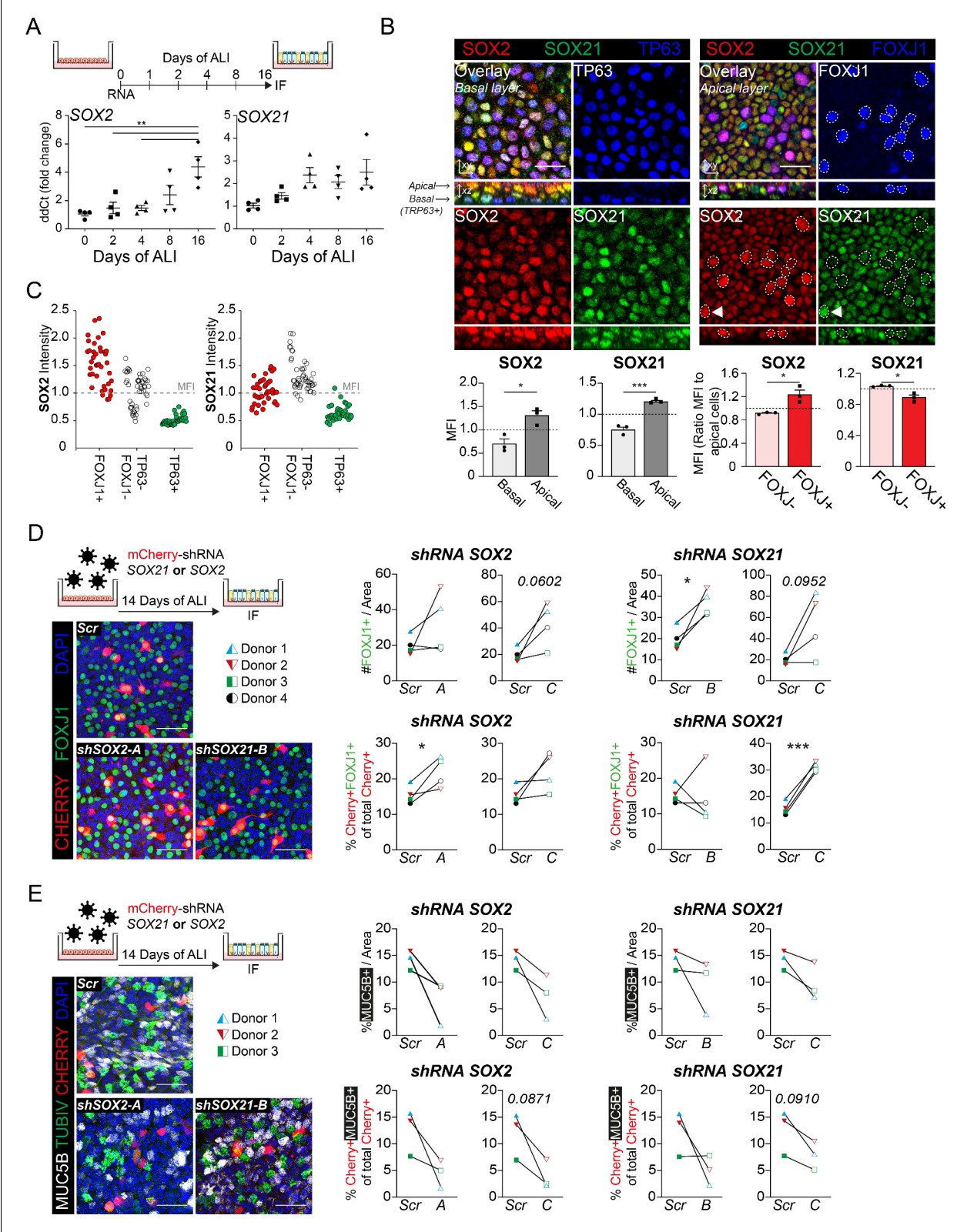

**Figure 7.** SOX2 and SOX21 in human basal cell differentiation. (**A**) Schematic overview of human primary bronchial epithelial cell (HPBEC) culture. QPCR analysis of *SOX2* and *SOX21* expression during differentiation of HPBECs on air–liquid Interface (ALI). One-way ANOVA (n = 4 (ALI cultures of four different donors); ***p<0.001). (**B**) Immunofluorescence analysis of HPBEC 16 days after ALI of SOX2 (red), SOX21 (green), and TP63 (blue; left) or FOXJ1 (blue; right). XZ plane shows basal cells (TP63+) at the basal membrane and ciliated cells (FOXJ1+) at the apical side. Scale bar = 25 μm. The

*Figure 7 continued on next page*

*Figure 7 continued*

grey bar graphs the MFI of SOX2 or SOX21 in basal and apical cells. The red bar graphs show the MFI of SOX2 or SOX21 in FOXJ+ and FOXJ− apical cells. The circled FOXJ1+ nuclei show high expression of SOX2 with sporadic high expression of SOX21(t) as well. Paired T-test (n = 3 [ALI cultures of three different donors]; *p<0.05, ***p<0.001). (**C**) Dot-plot of the MFI of SOX2 (y-axis: left graph) or SOX21 (y-axis: right graph). Each dot represents a cell, and ALI cultures of two different donors are included. The red filled circles are FOXJ1+-ciliated cells, green filled circles are TP63+ basal cells and non-filled are FOXJ1− TP63− cells. (**D**) Knockdown of *SOX2* or *SOX21* after lentiviral transduction of one of two different shRNAs directed against different regions of the *SOX2* or *SOX21* gene. Scr are HPBECs transduced with lentivirus and expressing a scrambled shRNA. The HPBEC ALI culture was transduced 1 day before exposure to air. The number of FOXJ1+ nuclei were analyzed after 14 days of ALI. Immunofluorescence of FOXJ1+ (green) ciliated cells and mCherry+ (red) transduced cells. Scale bar = 25 μm. The graphs show quantification of the average number of FOXJ1+ cells per area of 375.5 μm$^2$ (Top) and the percentage of FOXJ1+mCherry+ cells of total mCherry+ cells (bottom). Paired T-test (n = 4 [four different donors]; *p<0.05, ***p<0.001). (**E**) Immunofluorescence of HPBECs transduced with lentiviruses to knockdown *SOX2* (shSOX2-B) or *SOX21* (right; shSOX21-B) stained with TUBIV+ (green) to visualize differentiated ciliated cells, MUC5B+ (grey) for goblet cells and mCherry+ (red) denotes transduced cells. Scr are HPBECs transduced with lentivirus and expressing a scrambled shRNA. Scale bar = 25 μm. The area of MUC5B+ was measured after 14 days of ALI. The graphs show quantification of the average MUC5B+ per area of 375.5 μm$^2$. n = 3 (three different donors), each treated with two different shRNA constructs.

The online version of this article includes the following source data and figure supplement(s) for figure 7:

**Source data 1.** xlsx.

**Figure supplement 1.** SOX21 in human airway epithelium.

However, when co-expressed with SOX2, SOX21 is present in the region where SOX2 progenitor cells differentiate into basal cells. SOX2 and SOX21 can physically interact (*Mallanna et al., 2010*), and preliminary data of a genome-wide chromatin immunoprecipitation of the human airway ALI cultures showed that SOX2 and SOX21 not only have unique binding sites, but also co-occupy the same genomic regions (*Figure 7—figure supplement 1H*). This supports our findings that SOX2 and SOX21 are involved in the same cellular differentiation processes.

Using *Sox21$^{+/-}$* and *Sox21$^{-/-}$* mice, we showed that the presence of SOX21 is important to suppress the differentiation of SOX2+ progenitor cells into basal and ciliated cells. Surprisingly, basal cell numbers increased upon deletion of SOX21, but also after ectopic expression of a doxycycline-inducible SOX21 using an SPC-promotor- or SCGB1A1-promotor-driven rtTA during development. This seemingly contradictory result may be caused by the difference in timing between the mouse models. The *Sox21$^{-/-}$* mice lack expression of Sox21 from the fertilization of the oocyte, whereas the transgenic overexpressing mice only start to express SOX21 when the SPC or SCGB1A1 promoters become active. Thus, the *Sox21$^{-/-}$* mice lack Sox21 expression from the onset of lung development, which may trigger an expansion of basal cells, whereas the overexpression of SOX21 is induced in differentiated epithelial cells. The SPC or SCGB1A1 promoter-driven expression of SOX21 may cause a reversal of differentiated cells to basal cells. Alternatively, high levels of SOX21 may force cells to lose their identity followed by redifferentiation according to downstream cues received from the surrounding mesenchyme. For example, SPC-promoter-driven induction of SOX21 initiates the reversal of SOX9+ cells to a proximal phenotype, but this only becomes apparent once these SOX9+ cells are sufficiently distanced from distal mesenchymal FGF10 signaling. The latter causes the cells to express SOX2, leading to a more pronounced induction of the proximal fate and an increase in TRP63+ basal cells. The initiation of a proximal cell fate by ectopic expression of SOX21, followed by SOX2 expression, may explain why the cysts in SOX21-expressing lungs are smaller than the cysts observed upon ectopic expression of SOX2, the major proximal cell fate inducer.

SOX21 and SOX2 co-expression continues in adult TE and SMGs, which are both regions where progenitor cells reside. We used naphthalene injury and in vitro analysis to study whether SOX21 and SOX2 function in adult progenitors is similar to function during development. We showed that reduced SOX2 levels inhibit the contribution of SOX9+ SMG cells to repair of TE injury, while reduced SOX21 levels promote their contribution. Furthermore, reduced SOX2 levels decreased differentiation of basal to ciliated cells in vitro and in vivo, while reduced SOX21 levels only increased basal cell differentiation in vitro but not in vivo. The latter might be explained by the fact that differentiation of progenitor cells in vivo is regulated by an epithelial–mesenchymal interaction, which might inhibit further differentiation to ciliated cells when regeneration is complete (*Volckaert et al., 2017*). In summary, both during development and regeneration of the airway epithelium,

endogenous SOX21 acts as a suppressor of SOX2+ progenitor cell differentiation, while SOX2 levels are important for stimulating differentiation. Using a luciferase assay, we specifically show that SOX21 can antagonize SOX2 function on certain promotor regions. However, to fully understand the function of SOX21 within SOX2+ airway progenitors, signaling/environmental cues, additional co-factors, and expression levels are likely to play a role.

SOX2 and SOX21 showed a similar but not identical expression pattern in human airway epithelium. As opposed to mouse tracheal epithelium which has a basal and luminal cell layer, the human proximal airway epithelium contains an additional layer of intermediate cells or para-basal cells (*Mercer et al., 1994*; *Boers et al., 1998*). Using human primary bronchial epithelial ALI cultures, we showed high SOX21 levels in TP63− FOXJ1− cells and suggest that these cells represent the intermediate layer of para-basal cells transitioning to a luminal cell fate. Besides the different expression pattern compared to mice, SOX2 and SOX21 also act differently on basal cell differentiation, as reduced expression levels of either protein resulted in more ciliated cells and fewer goblet cells. In mouse ALI cultures, reduced SOX2 expression resulted in a decrease in ciliated cells, whereas reduced SOX21 expression resulted in an increase in ciliated cells without affecting secretory cell numbers. Recently, human basal cells were shown to comprise at least four basal cell types, whereas mice only have one basal cell population (*Travaglini et al., 2020*). For future studies, it is important the understand which genes are regulated by SOX2 and SOX21 in human and mouse, and how they subsequently affect airway epithelial homeostasis and regeneration.

Our data provide a new understanding of proximal–distal patterning of the airways and the regulation of SOX2 progenitor cells within development and regeneration of airway epithelium. De-regulation of SOX2 and SOX21 expression levels can alter branching morphogenesis and differentiation of airway epithelium. We show that SOX21 can act as a suppressor of differentiation when SOX2 expression levels are high and when progenitor cells are prone to differentiate.

# Materials and methods

**Key resources table**

| Reagent type (species) or resource | Designation | Source or reference | Identifiers | Additional information |
|---|---|---|---|---|
| Genetic reagent (*M. musculus*) | Sox2-CreERT | PMID:21982232 | MGI:5295990 | *Sox2*[tm1(cre/ERT2)Hoch] The Jackson Laboratories Stock No: 017593 |
| Genetic reagent (*M. musculus*) | Sox21-KO | PMID:19470461 | MGI:2654070 | Dr. Stavros Malas |
| Genetic reagent (*M. musculus*) | iSox2 | PMID:18374910 | MGI:98364 | Dr. Robbert Rottier |
| Genetic reagent (*M. musculus*) | iSox21 | - | MGI:2654070 | Dr. Robbert Rottier |
| Genetic reagent (*M. musculus*) | SPC-rtTA | PMID:11874100 | MGI:109517 | Prof. Jeffrey Whitsett |
| Genetic reagent (*M. musculus*) | CC10-rtTA | PMID:10766812 | MGI:98919 | Prof. Jeffrey Whitsett |
| Antibody | Mouse anti-β-TUBULIN IV | BioGenex | MU178-UC | 1:100 |
| Antibody | Mouse anti-FOXJ1 | eBioscience | 14–9965 | 1:200 |
| Antibody | Rabbit anti-KRT5 | Biolegend | Poly19055 | 1:500 |
| Antibody | Rat anti-KRT8 | DSHB | TROMA-I | 1:100 |
| Antibody | Rabbit anti-MYC (Immunofluorescence) | Abcam | AB9106 | 1:500 |
| Antibody | Rabbit anti-SCGB1A1 | Abcam | AB40873 | 1:200 |
| Antibody | Mouse anti-SCGB3A1 | R and D systems | AF1790 | 1:200 |
| Antibody | Goat anti-SCGB3A2 | R and D systems | AF3465 | 1:500 |
| Antibody | Mouse anti-SMA | Neomarkers | MS-113-P1 | 1:500 |

*Continued on next page*

*Continued*

| Reagent type (species) or resource | Designation | Source or reference | Identifiers | Additional information |
|---|---|---|---|---|
| Antibody | Rabbit anti-SOX2 | Seven-Hills | WRAB-SOX2 | 1:500 |
| Antibody | Goat anti-SOX2 | Immune systems | GT15098 | 1:500 |
| Antibody | Goat anti-SOX21 | R and D systems | AF3538 | 1:100 (TSA: 9 min) |
| Antibody | Rabbit anti-SOX9 | Abcam | AB185230 | 1:500 |
| Antibody | Mouse anti-TRP63 | Abcam | AB735 | 1:100 |
| Antibody | Rabbit anti-TRP73 | Abcam | AB40658 | 1:100 |
| Antibody | Rabbit anti-FLAG | Sigma | F7425 | Western blot, 1:1000 |
| Antibody | Rabbit anti-MYC | Abcam | AB9106 | 1:1000 |
| Antibody | Rabbit anti-MUC5B | Sigma | HPA008246 | 1:500 |
| Antibody | Mouse anti-β -TUBULIN | Sigma | T8328 | 1:2000 |
| Antibody | Alexa Fluor 405, 488, 594 Donkey anti Goat IgG | Jackson ImmunoResearch | 705-475-147, 705-545-147, 705-585-147 | 1:500 |
| Antibody | Alexa Fluor 488, 594, 647 Donkey anti Mouse IgG | Jackson ImmunoResearch | 715-545-151, 715-585-151, 711-605-151 | 1:500 |
| Antibody | Alexa Fluor 488, 594, 647 Donkey anti Rabbit IgG | Jackson ImmunoResearch | 711-545-152, 711-585-151, 711-605-152 | 1:500 |
| Antibody | Alexa Fluor 488, 594 Donkey anti Rat IgG | Jackson ImmunoResearch | 712-545-150, 712-585-153 | 1:500 |
| Antibody | Donkey anti-Goat HRP conjugated | Jackson ImmunoResearch | 705-035-147 | 1:500 |

## Mice

All animal experimental protocols were approved by the animal welfare committee of the veterinary authorities of the Erasmus Medical Center. Mice were kept under standard conditions. Mouse strains *SPC-rtTA* and *CC10-rtTA* (gifts of Jeffrey Whitsett), *Sox2-CreERT* (Jackson Labs, stock number 017593), *SOX21-KO* (gift of Stavros Malas), and *pTT::myc-SOX2* (iSox2) have been described (*Gontan et al., 2008*; *Kiso et al., 2009*; *Arnold et al., 2011*). The iSox21 transgenic mouse line was developed by cloning a murine *Sox21 cDNA* with a N-terminal myc epitope in a modified pTRE-Tight (Clontech) vector (pTT::myc-sox21). Pronuclear microinjection of a linearized fragment was performed, and three independent lines were tested. To induce expression of *Sox2* or *Sox21* during lung development, the *pTT::myc-Sox2* or *pTT::mycSox2*1 mice were crossed with SPC-rtTA or CC10-rtTA mice, and doxycycline was administered to dams in the drinking water (2 mg/ml doxycycline, 5% sucrose) from gestational day 6.5 onwards during development or for 6 weeks to 2–3 months old mice. Wild-type animals were non-transgenic littermates.

## Naphthalene injury

Adult mice (~8–12 weeks) were injured with a single intraperitoneal injection of 300 mg/kg naphthalene. Naphthalene (Sigma; 184500) was freshly prepared and dissolved in corn oil (Sigma; C8267). Corn oil injection served as baseline control. Groups of mice were sacrificed 2, 5, and 20 days post-injury (DPI) (number of mice per group is indicated in each figure).

## Mouse tracheal epithelial cell culture

Mouse tracheal epithelial cell (MTEC) culture was performed as previously described (*Eenjes et al., 2018*). Briefly, MTECs were isolated from mice adult trachea and cultured in KSFM expansion medium (*Table 1*) on collagen-coated plastic (50 µg/cm$^2$ of rat tail collagen Type IV [SERVA, 47256.01] in 0.02N acetic acid [Sigma; 537020]). After expansion, $8 \times 10^4$ MTECs were plated per collagen-coated 12-well insert (Corning Inc, Corning, USA) in proliferation medium (*Table 1*) for ALI culture. When confluent, MTECs were exposed to air by removing proliferation medium and adding

MTEC differentiation medium (*Table 1*) to the lower chamber. MTECs were cultured in standard conditions, at 37°C in a humidified incubator with 5% $CO_2$.

## Human primary epithelial cell culture

Human primary airway epithelial cell (HPBEC) culture was performed as previously described (*Amatngalim et al., 2018*). Lung tissue was obtained from residual, tumor-free material obtained at lung resection surgery for lung cancer. The Medical Ethical Committee of the Erasmus MC Rotterdam granted permission for this study (METC 2012–512).

Briefly, cells were isolated from healthy bronchial tissue by incubation in 0.15% Protease XIV (Sigma; P5147) for 2 hr at 37°C. The inside of the bronchi was scraped in cold PBS (Sigma; D8537), and the obtained airway cells were centrifuged and resuspended in KSFM-HPBEC medium for expansion (*Table 1*). Culture plates were coated with 10 µg/ml human fibronectin (Millipore; FC010), 30 µg/ml BSA (Roche; 1073508600), and 10 µg/ml PureCol (Advanced Biomatrix; 5005-B) for 2 hr at 37°C. Upon confluency, HPBECs were frozen ($4 \times 10^5$ cells/vial) and stored for later use.

When used for ALI culture, cells were thawed and seeded in a coated 10 cm dish, grown until confluent in KSFM-HPBEC medium, trypsinized, and $8 \times 10^4$ of HPBECs were plated per 12-transwell insert (Corning Inc, Corning, USA). On inserts, the HPBECs were cultured in bronchial epithelial growth medium (ScienCell Research Laboratories, Carlsbad, USA; 3211). Basal bronchial epithelial growth medium was first diluted 1:1 with DMEM (Gibco; 41966). Next, 1× supplement and 1× pen/strep (Lonza; DE17-602e) were added to 500 ml BEGM. Retinoic acid (1 nM RA) was freshly added. When cells reached full confluency, the BEGM medium was removed from the apical chamber and only supplied to the basal chamber and freshly supplemented with 50 nM RA. The medium was changed every other day, and the apical chamber was rinsed with PBS. HPBECs were cultured in standard conditions, at 37°C in a humidified incubator with 5% $CO_2$.

## Human fetal organoid culture

The culture of human fetal lung (17 weeks of gestation) and adult airway organoids was performed as previously described (*Nikolić et al., 2017*; *Miller et al., 2018*).

Lung lobes were dissociated using dispase (Corning; 354235) on ice for 30 min. Tissue pieces were then incubated in 100% FBS on ice for 15 min, after which they were transferred to a 10% FBS solution in Advanced DMEM with Glutamax, pen/strep, and HEPES. Lung bud tips were separated from mesenchymal cells through repeated pipetting. Pieces of tissue were resuspended in 30 µl of basal membrane extract (BME type 2, Trevigon; 3533-010-02), transferred to a 48-well plate, and incubated at 37°C to solidify the BME. After 5 min, 300 µl of self-renewing fetal lung organoid medium (*Table 1*) was added (*Nikolić et al., 2017*). Medium was refreshed every 3–4 days. Every 2 weeks, organoids were split 1:3 to 1:6. The medium was aspirated and cold PBS was added to the well to re-solidify the BME. Organoids were disrupted using a 1000 µl tip with on top a 2 µl tip. The disrupted organoids were centrifuged at 300 g for 5 min, resuspended in BME, and re-plated. To differentiate fetal lung organoids to organoids containing solely airway epithelium, fetal lung organoids were split, resuspended in BME, and re-plated with human airway organoid medium (*Table 1*; *Sachs et al., 2019*). Organoids were grown under standard culture conditions (37°C, 5% $CO_2$).

## Immunofluorescence

### Tissue

Mouse embryonic lungs and human bronchial tissue were fixed overnight in 4% PFA (Sigma; 441244) at 4°C. Post-fixation, samples were washed with PBS, de-hydrated to 100% ethanol, transferred to xylene, and processed to paraffin wax for embedding.

Organoids were retrieved from the BME by adding cold PBS to the 48 wells. Organoids were centrifuged at 150 g for 5 min and fixed overnight in 4% PFA at 4°C. Post-fixation, organoids sink to the bottom of the tube. Centrifuging was avoided after fixation to keep the organoids intact. The organoids were embedded in 4% low-melting agarose. The organoids were washed in PBS for 30 min and manually de-hydrated by 50 min incubation steps in 50%, 70%, 85%, 95%, and two times 100% ethanol. Organoids were further processed by three times xylene for 20–30 min and washed three times for 20–30 min in 60°C warm paraffin to remove all traces of xylene. The organoids were

**Table 1.** Medium.

| KSFM-hPBEC medium | | | |
|---|---|---|---|
| **Reagent** | **Company** | **Cat. no.** | **Final concentration** |
| KSFM | Gibco | 17005034 | n/a |
| Penicillin/streptomycin | Lonza | DE17-602e | 100 U/ml 100 µg/ml |
| Bovine pituitary extract | Gibco | 13028014 | 0.03 mg/ml |
| Human EGF | Peprotech | 315–09 | 25 ng/ml |
| Isoproterenol | Sigma | I-6504 | 1 µM |
| **KSFM-MTEC expansion medium** | | | |
| **Reagent** | **Company** | **Cat. no.** | **Final Concentration** |
| KSFM | Gibco | 17005034 | n/a |
| Penicillin/streptomycin | Lonza | DE17-602e | 100 U/ml 100 µg/ml |
| Bovine pituitary extract | Gibco | 13028014 | 0.03 mg/ml |
| Mouse EGF | Peprotech | 315–09 | 25 ng/ml |
| Isoproterenol | Sigma | I-6504 | 1 µM |
| Rock Inhibitor (Y27632) | Axon MedChem | 1683 | 10 µM |
| DAPT | Axon MedChem | 1484 | 5 µM |
| **MTEC proliferation medium** | | | |
| **Reagent** | **Company** | **Cat. no.** | **Final concentration** |
| DMEM:F12 | Gibco | 1133032 | n/a |
| Penicillin/streptomycin | Lonza | DE17-602e | 100 U/ml 100 µg/ml |
| NaHCO$_3$ | Gibco | 25080094 | 0.03% (w/v) |
| Fetal calf serum | HyClone | SH30071.03 | 5% |
| L-Glutamine | Gibco | 25030081 | 1.5 mM |
| Insulin-transferin-selenium | Gibco | 41400045 | 1× |
| Cholera toxin | Sigma | C8052 | 0.1 µg/ml |
| Bovine pituitary extract | Gibco | 13028014 | 0.03 mg/ml |
| Mouse EGF | Peprotech | 315–09 | 25 ng/ml |
| Rock inhibitor (Y27632) | Axon MedChem | 1683 | 10 µM |
| Retinoic acid | Sigma | R2625 | 0.05 µM |
| **MTEC differentiation medium** | | | |
| **Reagent** | **Company** | **Cat. no.** | **Final concentration** |
| DMEM:F12 | Gibco | 1133032 | n/a |
| Penicillin/streptomycin | Lonza | DE17-602e | 100 U/ml 100 µg/ml |
| NaHCO$_3$ | Gibco | 25080094 | 0.03% (w/v) |
| Bovine serum albumin | Gibco | 15260037 | 0.1% (w/v) |
| L-Glutamine | Gibco | 25030081 | 1.5 mM |
| Insulin-transferin-selenium | Gibco | 41400045 | 1× |
| Cholera toxin | Sigma | C8052 | 0.025 µg/ml |
| Bovine pituitary extract | Gibco | 13028014 | 0.03 mg/ml |
| Mouse EGF | Peprotech | 315–09 | 5 ng/ml |
| Retinoic acid | Sigma | R2625 | 0.05 µM |
| **Human fetal organoid medium** | | | |
| **Reagent** | **Company** | **Cat. no.** | **Final concentration** |
| Advanced DMEM:F12 | Invitrogen | 12634–034 | n/a |
| R-Spondin | Peprotech | 120–38 | 500 ng/ml |
| Noggin | Peprotech | 120–10C | 100 ng/ml |

*Table 1 continued on next page*

| Fgf10 | Peprotech | 100–26 | 100 ng/ml |
|---|---|---|---|
| Fgf7 | Peprotech | 100–19 | 100 ng/ml |
| EGF | Peprotech | AF-100–15 | 50 ng/ml |
| CHIR99021 | Stem Cell Techn. | 72052 | 3 µM |
| SB431542 | Tocris | 1614 | 10 µM |
| B27 supplement (- VitA) | ThermoFisher | 12587–010 | 1× |
| N-Acetylcysteine | Sigma | A9165 | 1.25 mM |
| Glutamax 100× | Invitrogen | 12634–034 | 1× |
| N2 | ThermoFisher | 17502–048 | 1× |
| Hepes | Gibco | 15630–56 | 10 mM |
| Penicillin/streptomycin | Lonza | DE17-602e | 100 U/ml 100 µg/ml |
| Primocin | Invivogen | Ant-pm-1 | 50 µg/ml |
| **Human airway organoid medium** | | | |
| **Reagent** | **Company** | **Cat. no.** | **Final concentration** |
| Advanced DMEM:F12 | Invitrogen | 12634–034 | n/a |
| R-Spondin | Peprotech | 120–38 | 500 ng/ml |
| Noggin | Peprotech | 120–10C | 100 ng/ml |
| Fgf10 | Peprotech | 100–26 | 100 ng/ml |
| Fgf7 | Peprotech | 100–19 | 25 ng/ml |
| SB202190 | Sigma | S7067 | 500 nM |
| A83-01 | Tocris | 2939 | 500 nM |
| Y-27632 | Axon MedChem | 1683 | 5 µM |
| B27 supplement | Gibco | 17504–44 | 1× |
| N-Acetylcysteine | Sigma | A9165 | 1.25 mM |
| Nicotinamide | Sigma | N0636 | 5 mM |
| Glutamax 100× | Invitrogen | 12634–034 | 1× |
| Hepes | Gibco | 15630–56 | 10 mM |
| Penicillin/streptomycin | Lonza | DE17-602e | 100 U/ml 100 µg/ml |
| Primocin | Invivogen | Ant-pm-1 | 50 µg/ml |

placed in a mold and embedded in paraffin. Paraffin blocks were sectioned at 5 µm and dried overnight at 37°C.

Sections were deparaffinized by three times 3 min xylene washes, followed by rehydration in distilled water. Antigen retrieval was performed by boiling the slides in Tris–EDTA (10M Tris, 1M EDTA) buffer pH = 9.0 for 15 min. Slides were cooled down for 30 min and transferred to PBS. For SOX21 staining, the tyramide signal amplification (TSA) kit was used (Invitrogen, B40922, according to manufacturer's protocol). When using the TSA kit, a hydrogen peroxide (35%) blocking step was performed after boiling. Sections were blocked for 1 hr at room temperature (RT) in 3% BSA (Roche; 10735086001), 0.05% Tween (Sigma; P1379) in PBS. Primary antibodies (Key resources table) were diluted in blocking buffer and incubated with the sections overnight at 4°C. The next day, sections were washed three times for 5 min at RT in PBS with 0.05% Tween. Secondary antibodies (Key resources table) were added in blocking buffer and incubated for 2 hr at RT. DAPI (4′,6-diamidino-2-phenylindole) solution (BD Pharmingen, 564907, 1:4000) was added to the secondary antibodies for nuclear staining. After incubation, three times 5 min washes in PBS-0.05% Tween and one wash in PBS were performed, and sections were mounted using Mowiol reagent (for 100 ml: 2,4% m/v Mowiol (Sigma; 81381), 4,75% m/v glycerol, 12 % v/v Tris 0.2M pH = 8.5 in dH$_2$O till 100 ml). All sections were imaged on a Leica SP5 confocal microscope.

## ALI culture

Human or mouse ALI cultures were washed with PBS and fixed on inserts in 4% PFA at RT for 15 min. Inserts were then washed 3 times for 5 min in 0.3% TritonX (Simga; T8787) in PBS and blocked for 1 hr at RT in 5% normal donkey serum (NDS, Millipore; S30), 1% BSA 0.3% TritonX in PBS. Primary antibodies (Key resources table) were diluted in blocking buffer and incubated overnight at 4° C. The next day, inserts were three times rinsed with 0.03% TritonX in PBS followed by three washes for 10 min at RT in PBS with 0.03% TritonX. Secondary antibodies (Key resources table) were added in blocking buffer and incubated for 2 hr at RT. DAPI solution (BD Pharmingen, 564907, 1:2000) was added to the secondary antibodies. After incubation, inserts were three times rinsed with 0.03% TritonX in PBS followed by three washes for 10 min at RT. Inserts were covered by a coverslip using Miowol reagent. Images were collated on a Leica SP5 confocal microscope.

## Image analysis

### Fluorescence intensity measurements

Intensity measurements in MTEC and HPBEC cultures were performed on three separate isolations from wild-type mice or donors and measured using ImageJ. Of each n, more that 500 nuclei were manually selected on the DAPI staining. In each nucleus, the intensity of SOX2 and SOX21 was measured. The MFI for each n and each intensity measurement was calculated by dividing it by the average intensity of that measurement in the same n.

For the knockdown ALI HPBEC culture, SOX2 and SOX21 intensity was measured in each experiment of each shRNA in 20 mCherry+ and 20 mCherry− nuclei.

### Counting

To standardize counting between animals, basal and ciliated cells were counted during lung development in a square of 400 μm around the first branch at the medial side of the bronchi. If counted on the lateral side, this is mentioned in the figure. In this manner, we could determine a position in the *SOX21*$^{-/-}$ animal where in the wildtype SOX21 is highest expressed. Of each genotype and each n, three sections were counted and the percentage of ciliated and basal cells were calculated based on the total number of airway epithelial (SOX2+) cells.

Five days after Naphthalene injury, the number of basal (TRP63+), ciliated (FOXJ1+), dividing (KI67+), secretory cells (SCGB1A1+), and SOX9+ cells were counted in the tracheal epithelium from cartilage ring (C) 0 till C1 for 5 dpi and from C0 till C6 for 20 dpi. Of each animal, three sections were counted throughout the trachea.

For the adult iSox2$^{CC10-rtTA}$ or iSox21$^{CC10-rtTA}$ experiments, the number of ciliated (FOXJ1+), secretory (SCGB1A1+), and basal (TRP63+) were counted in the trachea around cartilage ring (C) 1, C6, and C10. Of each animal, one section was counted.

The number of FOXJ1+ nuclei in the MTEC cultures were counted per 775 μm$^2$. For determining, the differentiation to secretory cells, the percentage of SCGB3A2+ area per 775 μm$^2$ was measured. The number of TRP63+ basal and TRP73+ cells were counted, respectively, to the number of nuclei present in each field. Each n are separate isolations of different animal, and per n, 5 fields of 775 μm$^2$ were counted.

ALI HPBEC cultures of four different donors were transduced with viruses to express shRNAs against *Sox2* or *Sox21*. Of each transduced culture, the number of FOXJ1+ nuclei and Cherry+ nuclei on three fields of 375.5 μm$^2$ was counted. The percentage of positive area MUC5B, TUBIV, and Cherry were measured on three fields of 375.5 μm$^2$ per insert.

## RNA isolation, cDNA synthesis, and qRT-PCR analysis

Human or mouse airway cells were removed from the insert by scraping them off the insert into cold PBS. Cells were collected in an Eppendorf tube and centrifuged at 800 g for 5 min at 4°C. PBS was aspirated, and the cell pellet was snap frozen in liquid nitrogen and stored at −80°C till RNA isolation.

To isolate RNA, 500 μl of TRI Reagent (Sigma, T9424) was added to the cell pellet. RNA extraction was performed according to the TRI Reagent protocol. RNA concentrations were measured using NanoDrop (ThermoFisher Scientific). First strand cDNA was synthesized using 2 μg RNA, MLV Reverse transcriptase (Sigma, M1302), and Oligo-dT primers (self-designed: 23xT+A, 23xT+C, and

23xT+G). For one qRT–PCR, 0.5 µl of cDNA was used with Platinum Taq polymerase (Invitrogen, 18038042) and SybrGreen (Sigma, S9430). The primer combinations for the qRT-PCR are listed in *Table 2*. Normalized gene expression was calculated using the ddCT method relative to GAPDH (mouse) or B-ACTIN (human) control.

## Luciferase assay
### Cloning
Promotor regions were PCR amplified (primers listed in *Table 3*) from mouse genomic DNA. To each primer, a restriction site was added, to clone the promotor region into the pGL4.10 [luc2] construct (Promega; E6651). The promotor sequence included the sequence of a transcriptional active area, which was identified with a RNA polymerase II ChIP in MTECs (*Marshall et al., 2016*). We used in silico analysis to predict SOX21 binding sites (MatInsepctor, GenomatixSuite v3.10 and http://jaspar2016.genereg.net/).

### Luciferase assay
HELA cells were plated in a 12-well plate in DMEM + 10% FCS and transiently transfected using Lipofectamine3000 (Thermofisher, L3000001). Each transfection consisted of 250 ng expression plasmid pcDNA3-control (Addgene; n/a), pcDNA3-Sox2FLAG (homemade), pcDNA3-Sox21MYC (homemade), 250 ng pGL4.10[luc2] reporter plasmid, and 2.5 ng TK-Renilla plasmid (Promega; E2241) (transfection control). Luciferase activity was measured 48 hr after transfection using the Dual-Luciferase Reporter Assay System (Promega, E1910). Plate reader VICTOR X4 was used to measure Firefly and Renilla luminescence. Firefly luminescence of each sample was calculated by dividing the Firefly luminescence by the Renilla luminescence. The increase or decrease of luciferase activity was then normalized to the pGL4.10[luc2] reporter plasmid transfected with the pcDNA3-control expression plasmid.

### Western blot
Samples were prepared from cell lysates used for luciferase measurements. Cells were lysed in the lysis buffer that was included in the Dual-Luciferase Reporter Assay System (Promega, E1910). To the cell lysate, 8M urea (to denature the DNA), 50 mM 1,4-dithiothreitol (DTT, Sigma), and 1× SDS sample buffer was added. Samples were boiled and loaded on a 12% SDS-polyacrylamide gel and blotted onto a PVDF membrane (Immobilon-P transfer membrane, Millipore). The blots were

**Table 2.** RT-PCR primers.

**RT-primers**

| Gene | Forward (5'→3') | Reverse (5'→3') | Species |
|---|---|---|---|
| *Foxj1* | CAGACCCCACCTGGCAGAATTC | AAAGGCAGGGTGGATGTGGACT | Mouse |
| *Gapdh* (housekeeping) | CCTGCCAAGTATGATGACAT | GTCCTCAGTGTAGCCCAAG | Mouse |
| *Krt5* | TACCAGACCAAGTATGAGGAG | TGGATCATTCGGTTCATCTCAG | Mouse |
| *Scgb1a1* | GCAGCTCAGCTTCTTCGGACA | TCCTGGTCTCTTGTGGGAGGG | Mouse |
| *Scgb3a2* | GTGGTTATTCTGCCACTGCCCTT | TCGTCCACACACTTCTTCAGTCC | Mouse |
| *Sox2* | AACATGGCAATCAAATGTC | TTGCCAGTACTTGCTCTCAT | Mouse |
| *Sox21* | TTGAAAGATGCCTCTCACCA | AATAAGCTAAATGGGAAGGGAG | Mouse |
| *Trp63* | GGAAAACAATGCCCAGACTC | GATGGAGAGAGGGCATCAAA | Mouse |
| *Actin* (housekeeping) | ATTGGCAATGAGCGGTTC | GGATGCCACAGGACTCCAT | Human |
| *Foxj1* | CCCACCTGGCAGAATTCAATCCG | CAGTCGCCGCTTCTTGAAAGC | Human |
| *Scgb1a1* | GCTCCGCTTCTGCAGAGATCTG | GCTTTTGGGGGAGGGTGTCCA | Human |
| *Scgb3a1* | TGCTGGGGGCCCTGACA | ACGTTTATTGAGAGGGGCCGG | Human |
| *Sox2* | AATGCCTTCATGGTGTGGTC | TTGCTGATCTCCGAGTTGTG | Human |
| *Sox21* | CCACTCGCTTGGATTTCTGACACA | TCGACTCAAACTTAGGGCAACGA | Human |
| *Trp63* | CCACCTGGACGTATTCCACTG | TCGAATCAAATGACTAGGAGGGG | Human |

**Table 3.** Primers used for pGL4.10[luc2] cloning.

**Primers used for pGL4.10[luc2] cloning**

| Name | Primers (5'→3') cut site | Cut site |
|------|--------------------------|----------|
| Trp63 | Forward: CAGGGTACCGGGCACATTCCATCTTTCCT | KpnI |
| | Reverse: CAGCTCGAGAGACTGGTCAAGGCTGCTCT | Xho |
| Sox2 | Forward: CAGGGTACCCGCGAGAGTATTGCAGGGAA | KpnI |
| | Reverse: CAGGCTAGCCGGAGATCTGGCGGAGAATA | NheI |
| Trp73 | Forward: CAGGGTACCGGACACGCATCTGTTGTGGA | KpnI |
| | Reverse: CAGCTCGAGTCTGCACACGCTGAGGAGCT | Xho |

blocked for 1 hr in PBS containing 0.05% Tween-20% and 3% BSA at RT, and probed overnight with primary antibodies at 4°C (Key resources table). Next day, membranes were washed three times with PBS containing 0.05% Tween-20 and incubated for 1 hr with horseradish peroxidase (HRP)-conjugated secondary antibodies (DAKO) at a dilution of 1:10,000. Signal was detected with Amersham ECL Prime Western Blotting Detection Reagent (GE Healthcare). Blots were developed using the Amersham Imager 600GE (GE Healthcare).

## Knockdown of *SOX2* and *SOX21* in HPBEC ALI cultures

### Lentiviral production

Lentiviral constructs containing an U6 promoter to express shRNAs targeted at *SOX2* or *SOX21* were ordered at Genecopoeia (HSH110118-LVRU6MP-a to c, HSH091508-LVRU6MP-a to c). Three constructs targeting different parts of *SOX2* or *SOX21* were used, and a non-targeting scrambled shRNA construct was used as negative control (CSHCTR001-1-LVRU6MP; referred to as shScrm). To visualize efficient transduction into HPBECs, all plasmids contained a mCherry expression cassette to visualize transduced cells.

Lentivirus was generated by transfection of HEK293T cells. A 10 cm dish of HEK293T cells was transfected using 100 μg PEI with 20 μg shRNA plasmid, 15 μg Pax2 (packaging vector), and 5 μg VSV-G (envelope vector). After 4 hr, medium was replaced with 8 ml DMEM + 10% FCS + 1× pen/strep. Medium was collected and refreshed after 24 hr, 48 hr, and 72 hr. The collected medium was centrifuged at 1500 rpm for 5 min, and supernatant was filtered through a 0.45 μm filter to discard all remaining cells. To collect lentiviral particles, the supernatant was centrifuged at 20,000 rpm, at 4°C for 135 min. Virus pellets were resuspended in 100 μl BEGM medium.

### Knockdown of SOX2 or SOX21 in HPBEC ALI cultures

$1 \times 10^5$ HPBECs were seeded onto 12 mm inserts, 0.4 μm pore (Corning Inc, Corning, USA), and grown submerged until confluent. The HPBECs were transduced with 10–20 μl lentiviral particles in 500 μl BEGM + 1 nM EC23 (retinoic acid analog, Sigma; SML2404) + 2 μg/ml Polybrene in the apical compartment of the insert. After 24 hr, the apical compartment was washed once with PBS and the PBECs were put to ALI. Medium was changed in the lower chamber to BEGM + 50 nM EC23. Transduced HPBECs were cultured on ALI for 14 days, after which they were fixed and processed for immunofluorescence staining.

## Statistics

Statistical analysis was performed using Prism5 (GraphPad). For all measurement, three or more biological replicates were used. Data are represented as means ± standard error of mean (SEM) with the data points present in each graph. Statistical differences between samples were assed with Student's unpaired or paired t-test, one-way ANOVA (post-test: Tukey), or two-way ANOVA (post-test: Bonferroni). p-values below 0.05 are considered significant. The number of replicates and statistical tests used are indicated in the figure legends.

## Single-cell RNA sequencing of HPBEC

Expression levels of *SOX2* and *SOX21* in ALI cultures of HPBEC were analyzed using the dataset previously published (*Plasschaert et al., 2018*). The dataset is available at the Klein Lab SPRING viewer.

## Acknowledgements

We thank, Mart Lamers and Bart Haagmans (Department of Viroscience, Erasmus MC) for assistance and supplying the human fetal lung organoids, Thomas Koudstaal (department of Pulmonary Medicine, Erasmus MC) for supplying lung resection material for the isolation of human primary bronchial epithelial cells, Rutger Brouwer (Department of Biomics, Erasmus MC) for Figure 7-supplement 6H and Frank Grosveld for critically reading the manuscript (Department of Cell Biology, Erasmus MC). This work was supported by grants from the Sophia Foundation for Medical Research (grant numbers S14-12 EE; S17-20 GGE; S16-17 JJPC).

## Additional information

### Funding

| Funder | Grant reference number | Author |
| --- | --- | --- |
| Sophia Foundation for Medical Research | S14-12 | Evelien Eenjes |
| Sophia Foundation for Medical Research | S17-20 | Gabriela G Edel |
| Sophia Foundation for Medical Research | S16-17 | Jennifer Collins |

The funders had no role in study design, data collection and interpretation, or the decision to submit the work for publication.

### Author contributions

Evelien Eenjes, Conceptualization, Resources, Data curation, Software, Formal analysis, Validation, Investigation, Visualization, Methodology, Writing - original draft, Writing - review and editing; Marjon Buscop-van Kempen, Anne Boerema-de Munck, Software, Formal analysis, Investigation, Methodology; Gabriela G Edel, Data curation, Formal analysis, Investigation, Methodology; Floor Benthem, Lisette de Kreij-de Bruin, Formal analysis, Investigation; Marco Schnater, Jennifer Collins, Validation, Writing - review and editing; Dick Tibboel, Funding acquisition, Writing - review and editing; Robbert J Rottier, Conceptualization, Resources, Supervision, Funding acquisition, Validation, Investigation, Methodology, Writing - original draft, Project administration, Writing - review and editing

### Author ORCIDs

Robbert J Rottier ![ORCID] https://orcid.org/0000-0002-9291-4971

### Ethics

Animal experimentation: All animal experimental protocols were approved the national committee ("Centrale Commissie Dierproeven"; number AVD101002017871) and by the animal welfare committee of the veterinary authorities (prtocol numbers 17-871-01 and 17-871-02) of the Erasmus Medical Center.

### Decision letter and Author response

Decision letter https://doi.org/10.7554/eLife.57325.sa1
Author response https://doi.org/10.7554/eLife.57325.sa2

## Additional files

### Supplementary files
- Transparent reporting form

### Data availability
All data generated or analysed during this study are included in the manuscript and supporting files. Source data files have been provided for Figures 1 to 5, and 7.

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
