## [Decision Letter]

**Acceptance summary:**

This study aims at advancing our understanding about the molecular mechanisms that control airway stem cell maintenance and differentiation. The analysis focuses on Sox21 in lung development, homeostasis, and injury, using wide array of methods including mouse human cell-based ALI cultures and organoids. The authors describe a new role for Sox21 downstream of the key airway development transcription factor *Sox2*. They suggest opposing roles for these genes whereby Sox21 prevents and *Sox2* promotes differentiation of immature airway progenitors into basal cells. This study thus describes a role of Sox21 in the (patho)biology of airway progenitors and the balance of cell types in the airways.

**Decision letter after peer review:**

Thank you for submitting your article "SOX21 modulates *SOX2*-initiated differentiation of epithelial cells in the extrapulmonary airways" for consideration by *eLife*. Your article has been reviewed by 3 peer reviewers, one of whom is a member of our Board of Reviewing Editors, and the evaluation has been overseen by Edward Morrisey as the Senior Editor. The following individuals involved in review of your submission have agreed to reveal their identity: Stijn De Langhe (Reviewer #2); Finn Hawkins (Reviewer #3).

The reviewers have discussed the reviews with one another and the Reviewing Editor has drafted this decision to help you prepare a revised submission.

Summary:

This is a comprehensive analysis of Sox21 in lung development, homeostasis, and injury, using wide array of methods including mouse human cell-based ALI cultures and organoids. The authors describe a new role for Sox21 downstream of the key airway development transcription factor *Sox2*. They suggest opposing roles for these genes based on analysis of *Sox2* and Sox21 heterozygous knock-out mice whereby Sox21 prevents and *Sox2* promotes differentiation of immature airway progenitors into basal cells. This raises interesting questions regarding the role of Sox21 in the biology of airway progenitors and the balance of cell types in the airways. In adult mice no obvious differences are detected in Sox21+/- mice in terms of regenerative capacity. In in vitro assays of adult mouse basal cell differentiation (ALI culture) failed to identify obvious differences in basal cell number of differentiation into specific lineages in Sox21+/- cells. In human ALI cultures the authors identify a similar possible role for Sox21 in differentiating basal cells in humans. Overall, the molecular mechanisms that control airway stem cell maintenance and differentiation are of great interest to the field. However, in its current form the studies remain primarily descriptive and several data inconsistencies remain, which makes it difficult to draw impactful conclusions at the current stage. The potential suitability for *eLife* would significantly increase with additional mechanistic experiments and data.

Essential revisions:

In particular, the following major points require additional experimental data (details are outlined below):

1) The functional Sox21 role in the adult lung is not clear.

– Mouse studies would benefit from inclusion of overexpression of Sox21 in the airways.

– In human airways and the models used (organoids and ALI cultures) gain and loss of function studies are not performed and would also behelpful to further determine the phenotype of the intermediate para-basal cells in the human airway.

2) Evidence for genetic interaction between Sox21 and *Sox2* in the adult lung needs to be expanded.

3) Presence of *Sox9* positive basal cells as additional atypical basal cells needs to be further investigated.

4) Quantification the % of club cells in the corn oil and naphthalene treated tracheas of the different mutants as these are the main cells that should be replaced upon naphthalene injury.

*Reviewer #2:*

This is an interesting but also a bit confusing manuscript by Evelien Eenjes and colleagues.

I was wondering whether the seemingly conflicting effect of Sox21 on basal cell differentiation could be due to an increase or decrease in *Sox9* positive basal cells. These atypical basal cells have been described in a number of manuscripts. For example https://www.ncbi.nlm.nih.gov/pubmed/26869074.

It appears from the figures that Sox21 may increase *Sox9* expression. Can the authors look at the presence of *Sox9* positive basal cells in the trachea of the different mutants?

It would be important to quantify the % of club cells in the corn oil and naphthalene treated tracheas of the different mutants as these are the main cells that should be replaced upon naphthalene injury. I am not sure that people normally look at the number of ciliated cells reappearing after naphthalene injury as ciliated cells are not thought to be killed by naphthalene.

The authors should also look at *Sox9* positive basal cells in the tracheas of the corn oil and naphthalene treated ctrl and mutant mice.

Are the intermediate para-basal cells in the human airway differentiation experiment club cells? Why do the authors think these cells are on their way to become ciliated cells? Doesn't Sox21 inhibit ciliated cell differentiation?

*Reviewer #3:*

In their manuscript titled "SOX21 modulates *SOX2*-initiated differentiation of epithelial cells in the extrapulmonary airways" from Eenjes et al. the authors describe a new role for Sox21 in the developing and adult airway epithelium. Building on prior work they observe a unique distribution of Sox21 expression in the developing airways and through careful and elegant immunostaining and over-expression studies they suggest that Sox21 is downstream of a key airway development transcription factor *Sox2*. They suggest opposing roles for these genes based on analysis of *Sox2* and Sox21 heterozygous knock-out mice whereby Sox21 prevents and *Sox2* promotes differentiation of immature airway progenitors into basal cells. This raises interesting questions regarding the role of Sox21 in the biology of airway progenitors and the balance of cell types in the airways. In adult mice no obvious differences are detected in Sox21+/- mice in terms of regenerative capacity. In in vitro assays of adult mouse basal cell differentiation (ALI culture) failed to identify obvious differences in basal cell number of differentiation into specific lineages in Sox21+/- cells. The authors report an overall similar pattern of SOX21 expression in a human fetal lung organoid platform. Finally, in human ALI cultures the authors identify SOX21 expression most abundant in paranasal cells suggesting a similar possible role for this genes in differentiating basal cells in humans. Overall, the molecular mechanisms that control airway stem cell maintenance and differentiation are of great interest to the field.

The development data is clear based on the studies presented. The immunostaining and figures are elegant. It does not appear the Sox21 is necessary for airway development. A key question raised by this work is how important and what precisely is the role of Sox21? For example, overexpression of Sox21 in the developing and adult airways might have been more instructive in answering these key questions rather than under the control of the SPC-promoter.

The adult mouse data and human data seem to overall suggest that *SOX2* and SOX21 are upregulated in differentiating cells in vitro and are expressed at lower levels in basal cells but real mechanism or phenotype related to SOX21 function is observed in the SOX21 het mouse cells.

The attempt to assess human fetal airways is admirable. However, sections of fetal lungs would be more relevant. Can the authors address how similar fetal lung cultures are to in vivo airways and at what developmental time point ?

Overall the relevance of SOX21 expression in the adult airways is unclear.

[Editors' note: further revisions were suggested prior to acceptance, as described below.]

Thank you for submitting your article "SOX21 modulates *SOX2*-initiated differentiation of epithelial cells in the extrapulmonary airways" for consideration by *eLife*. Your article has been reviewed by 3 peer reviewers, one of whom is a member of our Board of Reviewing Editors, and the evaluation has been overseen by Edward Morrisey as the Senior Editor. The following individual involved in review of your submission has agreed to reveal their identity: Finn Hawkins (Reviewer #3).

Essential revisions:

The manuscript improved significantly several additional experiments included that have been ask for in the first revision. These experiments further clarified and support the dynamic and context-dependent role for Sox21 in airway cell differentiation. No further experimental data are required, however, some textual changes to clarify the results and enhance readability.

1) The manuscript is still very hard to read, especially given the variety in model systems and different results based on these. Please carefully revise and modify the text accordingly to clarify the results and enhance readability.

2) Adding 1-2 small figures to explain / visualize the experimental setups/ findings would also be helpful.

*Reviewer #1:*

The manuscript has significantly improved. Additional experiments have been performed including in vivo over expression of Sox21 and Sox 2 (Figure 4) as well as loss of function studies for Sox21 in Figure 7. These data further support a role for Sox21 in inhibiting differentiation of airway progenitors and adult basal cells. Specifically, airway cell transitioning from basal cells to a luminal para-basal cell while cilia cell transition/maturation is suppressed by Sox21. The manuscript is still very hard to read and textual clarification is needed.

*Reviewer #2:*

This article is very difficult to read. As currently written I do not understand it.

*Reviewer #3:*

This manuscript is of relevance to research into lung development and the mechanisms that relegate the maintenance and differentiation of progenitor cells in the airway epithelium.

The authors include several new experiments in response to concerns raised in the initial submission. The resubmission is significantly revised.

To address the role of SOX21 in airways in development and in adult mice under homeostatic conditions the authors over-expressed SOX21under the control of the SCGB1A1 promoter in mice using an inducible system. Over-expression during development led to an increase in basal cells. Over-expression of SOX21 in adult secretory cells led to fewer secretory cells not did not alter basal cell numbers. Interestingly, in both experiments the majority of cell over expressing the MYC-tag were not clearly secretory, basal or ciliated cells. While the experiments are helpful in evaluating the functional role of SOX21 in the airways the latter observation is troubling and not really addressed.

Concern in initial submission: The adult mouse data and human data seem to overall suggest that *SOX2* and SOX21 are upregulated in differentiating cells in vitro and are expressed at lower levels in basal cells but (no) real mechanism or phenotype related to SOX21 function is observed in the SOX21 het mouse cells. In response the authors performed knockdown of SOX21 and *SOX2* in human ALI cultures which suggest a role for both genes in suppressing the differentiation of basal cells to ciliated cells. While an increase in FOXJ1 was detected in these knockdown experiments, the authors did not detect an increase in multiciliated cells.

In addition to assessment of SOX21 in human fetal lung organoids sections of human fetal lung and adult human lung are now included.

Overall, the authors have performed several experiments that have improved the manuscript. It appears that the role of SOX21 is context dependent and despite the authors detailed experiments the biology remains complicated to interpret.

---

## [Author Response]

Essential revisions:In particular, the following major points require additional experimental data (details are outlined below):1) The functional Sox21 role in the adult lung is not clear.– Mouse studies would benefit from inclusion of overexpression of Sox21 in the airways.

We added the analysis of the effects of overexpressing SOX21 (and *SOX2*) on the epithelial differentiation of the airways late in gestation at E18.5 in figure 3E. Moreover, we studied the effect of SOX21 (and *SOX2*) overexpression during adult homeostasis using additional mouse experiments, and added figure 4E and figure 4—figure supplement 1E to the manuscript to discuss the results.

– In human airways and the models used (organoids and ALI cultures) gain and loss of function studies are not performed and would also behelpful to further determine the phenotype of the intermediate para-basal cells in the human airway.

We have performed *SOX2* and SOX21 knock down experiments in our human Air Liquid Interface (ALI) cultures and added the data to figure 7 and figure7-supplement 1.

2) Evidence for genetic interaction between Sox21 and Sox2 in the adult lung needs to be expanded.

For the genetic interaction, we have tried to obtain*Sox2*^+/-^Sox21^-/-^ mice, but the main difficulty resides in the fact that it is nearly impossible to breed these mice, and we did not obtain (sufficient) mice to perform reasonable experiment. We did analyze *Sox2*^+/-^, Sox21^+/-^, and Sox21^-/-^ E18.5 lungs, which we added to the manuscript (figure 3E, supplemental figure 1E). We obtained only one *Sox2*^+/-^Sox21^+/-^ mouse pup at E18.5, of which the lungs were analyzed. Since this is only one sample, we did not include it in the manuscript, but we like to share these preliminary data with the reviewer (see Author response image 1). It appears that the *Sox2*^+/-^Sox21^+/-^ lung is comparable to the individual *Sox2* and Sox21 heterozygous lungs.

**Author response image 1. respfig1:** Bar graphs representing the percentage of FOXJ1+, TRP63+ and SCGB1A1+ cells in E18. 5 lungs of WT, *Sox2Sox2*^+^/-, Sox21+/-, Sox21-/- and *Sox2Sox2*^+^/-Sox21+/- mice.

Previously, we observed a direct interaction between *SOX2* and SOX21, which was already published by others (Mallanna et al., Stem Cells 28, 1715-1727, 2010). Thus, there is a direct link between the two proteins. Moreover, we performed *SOX2* and SOX21 ChIP-seq experiments using human ALI cultures. However, we will not report on the full data set (yet), but have included four genome tracks to show single *SOX2* or SOX21, and double *SOX2*-SOX21 occupancy at different genomic loci (Figure 7—figure supplement 1H). We included the supplemental figure, and refer to the data in the discussion line (452-457), to underly the importance of studying the interaction between these proteins further. Of note, the human ALI system is a culture method with a diverse range of cell types and all cell types are in different status of differentiation. It is challenging to study the interaction and function(s) of *SOX2* and SOX21 at the cellular level. As described previously and in the discussion (line 446-448), the interaction of both *SOX2* and SOX21 are highly context dependent, emphasizing the importance of obtaining a suitable system to study this interaction in the human and mouse airway epithelium.

3) Presence of Sox9 positive basal cells as additional atypical basal cells needs to be further investigated.

We have further investigated the putative presence of *SOX9*+ basal cells in our experiments, and found that there are some rare *SOX9*+ basal-like cells. To acknowledge these findings, we have adapted the text (line 178-184) and added supplemental figure2-supplement 2E. As *SOX9*+ basal cells lack expression of the KRT5 basal marker (Utisyan 2016), we added a KRT5 staining of *Sox2^+/-^, Sox21^+/-^* and *Sox21^-/-^* E18.5 lungs to analyze the number of basal cells (Figure 3E and figure 3 supplement 1E).

4) Quantification the % of club cells in the corn oil and naphthalene treated tracheas of the different mutants as these are the main cells that should be replaced upon naphthalene injury.

We have quantified the percentage of Club cells before and after the treatment with naphthalene, and we did not observe significant differences between the different genotypes, or between treated and non-treated animals. We have included these data in figure 5D

Reviewer #2:This is an interesting but also a bit confusing manuscript by Evelien Eenjes and colleagues.I was wondering whether the seemingly conflicting effect of Sox21 on basal cell differentiation could be due to an increase or decrease in Sox9 positive basal cells. These atypical basal cells have been described in a number of manuscripts. For example https://www.ncbi.nlm.nih.gov/pubmed/26869074.It appears from the figures that Sox21 may increase Sox9 expression. Can the authors look at the presence of Sox9 positive basal cells in the trachea of the different mutants?

The reviewer points out an interesting observation. In Figure 1C we indeed find that when *Sox21* is induced in the lung bud, *SOX9* remains expressed. Furthermore, in the airway we find some iSOX21 cells that are still low in expression of *SOX9*, but these are scarce and not all iSOX21 expressing cells do.

We did find the presence of TRP63 expressing *SOX9*+ basal cells in the SPC induced animals. However, we did not elaborate! As described by Ustiyan et al., the observed *SOX9*+ TRP63+ atypical basal cells are KRT5 negative. We included therefore KRT5 in our staining panel to analyze the number of basal cells in the proximal airways upon induction of *Sox2* or *Sox21* using the CC10 rTta (Figure 2B-E). All basal cells analyzed were also positive for KRT5. KRT5 was also added to analyze the number of basal cells in *Sox2^+/-^, Sox21^+/-^* and *Sox21^-/-^* mice at E18.5 (Figure 3E).

We have included these findings in the text and figures:

“Previously, conditional deletion of βcatenin in SPC+ cells during lung development showed the appearance of cystic structures, similar to the induction of *SOX2* and SOX21. […] Similarly we also found *SOX9*+ basal cells in the iSox2^SPC-rtTA^ and iSox21^SPC-rtTA^ mouse lungs, and sporadic *SOX9*+ basal cells were also detected in the control (Figure 2—figure supplement 1E).”

It would be important to quantify the % of club cells in the corn oil and naphthalene treated tracheas of the different mutants as these are the main cells that should be replaced upon naphthalene injury. I am not sure that people normally look at the number of ciliated cells reappearing after naphthalene injury as ciliated cells are not thought to be killed by naphthalene.

The reviewer has indeed a good point, and we have quantified the number of Club cells before and after treatment with naphthalene. We have adapted the figure and the text accordingly:

“To determine whether *SOX2* or SOX21 deficiency affects regeneration after naphthalene injury, we quantified the percentage of secretory, ciliated, non-dividing and dividing basal cells at 20 DPI. Fewer FOXJ1+ ciliated cells were observed in *Sox2*^+/-^ TE compared to WT, Sox21^+/-^ , or to *Sox2*^+/-^ CO exposed mice (Figure. 5D, Figure 5-supplement 1D).”

The authors should also look at Sox9 positive basal cells in the tracheas of the corn oil and naphthalene treated ctrl and mutant mice.

The *SOX9*+ basal cells is missing in the analysis and unfortunately were not able to add it to the naphthalene experiments, due to lack of material to quantify. However, we did stain TRP63 and *SOX9* at 5 days post injury and *SOX9*+ cells are present at the surface epithelium together with TRP63+ basal cells , we cannot exclude that these basal cells are *SOX9*+ and whether there is a difference. However, at day 5 post-injury we did not see a difference in the number of basal cells between genotypes but a small difference in *SOX9*+ cells.

Are the intermediate para-basal cells in the human airway differentiation experiment club cells? Why do the authors think these cells are on their way to become ciliated cells? Doesn't Sox21 inhibit ciliated cell differentiation?

The SOX21 high expressing cells in the ALI culture did not correlate with SCGB1A1 expression (Author response image 2, n=1). We did not continue these experiments, as the single cell RNA seq of human ALI cultures shows a high expression of *SOX21* in para-basal cells (Figure 7- figure supplement 1E) and the localization of high expressing SOX21 on human sections are found in the intermediate layer (Figure 6D). As there are some FOXJ1+ nuclei high in expression of SOX21 (Figure 7—figure supplement 1D) we interpreted the data as cells transition to ciliated cells. We observed with the added knock-down experiments that SOX21, similar as in mouse, inhibits ciliated cell differentiation in human ALI cultures. We excluded this possibly premature statement from the manuscript and now state the following:

“We hypothesize that these cells are intermediate cells transitioning from basal cells to a luminal cell fate, the para-basal cell. […] This analysis confirmed that high levels of SOX21 mRNA denote an intermediate cell type.”

**Author response image 2. respfig2:** Intensity of *SOX2* or SOX21 was measured in nuclei of SCGB3A1+ cells and SCGB3A1- cells. Circles were put around nuclei that were SCGB3A1+. Staining of SCGB3A1 was analyzed throughout the z-stack.

Reviewer #3:[…] The development data is clear based on the studies presented. The immunostaining and figures are elegant. It does not appear the Sox21 is necessary for airway development. A key question raised by this work is how important and what precisely is the role of Sox21? For example, overexpression of Sox21 in the developing and adult airways might have been more instructive in answering these key questions rather than under the control of the SPC-promoter.

The reviewer raises a good point, and indeed, the SOX21 knockout mice suggests that SOX21 is not necessary for lung development. However, SOX21 has a clear influence on the distribution of the different cell types, as we report.

Initially, we choose to use the SPC-driven rtTA, as this transgene is expressed early in development in most epithelial cells (e.g. Gontan et al., 2008). These data were included in the original manuscript. We have now expanded this data set using a mouse model to induce *SOX2* or SOX21 in differentiating epithelial cells, using the rtTA under the control of the CC10 promoter. This resulted in additional embryonic data, as well as new adult data, which we have included. For the embryonic data, we included the following text and accompanying figures:

“To further investigate the role of *SOX2* and SOX21 in differentiation of airway epithelium during development, we induced expression of the MYC-tagged *Sox2* or Sox21 transgenes in secretory cells using a CC10/SCGB1A1 driven rtTA inducible model (iSox2^CC10-rtTA^ and iSox21^CC10-rtTA^) (Tichelaar, Lu et al. 2000). […] So, ectopic expression of *SOX2* and SOX21 during development stimulates the appearance of basal cells and decrease the number of ciliated cells.”

And for the adult data set:

“To further understand the role for *SOX2* and SOX21 in adult airway epithelium, we used the iSox2^CC10-rtTA^ and iSox21^CC10-rtTA^ mice to induce expression of *Sox2* or Sox21 in secretory cells for 6 weeks (Figure 4E). […] Similar as during development, the MYC+ transgene expressing cells were mostly negative for other cell specific markers in both mouse models (Figure 4—figure supplement 1E).”

The adult mouse data and human data seem to overall suggest that SOX2 and SOX21 are upregulated in differentiating cells in vitro and are expressed at lower levels in basal cells but real mechanism or phenotype related to SOX21 function is observed in the SOX21 het mouse cells.

We are not sure what the exact question is of the reviewer. In light of this, we have now expanded the human data with knock down experiments using shRNA constructs in human ALI cultures (figure 7 and figure 7—figure supplement 1). These data show comparable effects of reduced *SOX2* or SOX21 as observed in the mouse cells.

The attempt to assess human fetal airways is admirable. However, sections of fetal lungs would be more relevant. Can the authors address how similar fetal lung cultures are to in vivo airways and at what developmental time point ?

We have performed immunofluorescence staining on human sections throughout gestation and included these in figure 6A. We have restructured the figures, so figure 6 now displays human fetal lung sections, organoids (fetal tip organoids and differentiated organoids), and adult bronchi. The ALI cultures we have used for the mechanistic experiments mostly resemble the adult bronchus.

Overall the relevance of SOX21 expression in the adult airways is unclear.

We have performed additional experiments to investigate the role of *SOX2* and SOX21 in the differentiating human epithelium by shRNA knockdown experiments, described in the text, figure 7 and supplemental figure 6. We have added the following text:

“Next, we assessed the effects of reduced levels of *SOX2* or SOX21 on human basal cell differentiation by transducing ALI cultures with lentiviruses expressing short hairpin RNAs (shRNA) to knockdown *SOX2* or SOX21. […] Overall, our data suggest that both *SOX2* and SOX21 suppress basal to ciliated cell differentiation, while both seem to have a positive role in the differentiation of goblet cells in the human airways.”

Combined with the mouse data (*Sox2*^+/-,^ Sox21^+/-^ and Sox21^-/-^; Naphthalene; CC10rTTA adult), we hope to better indicate the relevance of SOX21 in the adult airways.

[Editors' note: further revisions were suggested prior to acceptance, as described below.]

Essential revisions:The manuscript improved significantly several additional experiments included that have been ask for in the first revision. These experiments further clarified and support the dynamic and context-dependent role for Sox21 in airway cell differentiation. No further experimental data are required, however, some textual changes to clarify the results and enhance readability.1) The manuscript is still very hard to read, especially given the variety in model systems and different results based on these. Please carefully revise and modify the text accordingly to clarify the results and enhance readability.

We agree with the reviewers that the inclusion of new experimental data in the revised version resulted in loss of clarification of the some of our data, which regrettably led to reduced readability. We have carefully revised the new manuscript, and made several textual changes to enhance readability. To do so, we have consulted three independent researchers, including a native speaker, to better foreword our data, conclusions and discussion. We hope the reviewers agree that the manuscript has indeed been improved. Since we have made several changes, we have included a marked-up version to better highlight the changes to the manuscript.

2) Adding 1-2 small figures to explain / visualize the experimental setups/ findings would also be helpful.

We have included two new panels (figure 2F and figure 4F) to summarize and highlight our findings. We believe that these two extra cartoons indeed clarify the data in an appropriate manner, and thank the reviewers for this suggestion. Moreover, throughout the manuscript’s figures and figure supplements, we have several cartoons to explain our experimental setup (e.g. 2B, 4A, 4E, 5B, 7A), as well as three data-summarizing cartoons (Figures 3G, Figure 2-supplement 1D, Figure 4-supplement 1D).

In addition, we have made several (minor) changes to the figures to comply with the changes in the text:

Figure 1

Panel A: Add “Trachea” and “Carina”

Panel C: Add the duration of Dox treatment Figure 1—figure supplement 1:

Panel B: changed text color of DOX to red

Panel C: Line made red to show that the mice received doxycycline from E6 to E18

Figure 2

Panel B: Removed “1st branch” above cartoon, added “Carina”

Panel F: New scheme to illustrate the expression of MYC-*SOX2* or MYC-SOX21 in secretory cells during development.

Figure 4

Panel F.: Added a scheme to illustrate the expression of MYC-*SOX2* or MYC-SOX21 in SCGB1A1+ secretory cells in adult tracheal epithelium.

Figure 7

Panel B: Changed “luminal” to “apical” Figure 7—figure supplement 1:

Panel C: Changed graph y-ax to apical.